# Sheep genome functional annotation reveals proximal regulatory elements contributed to the evolution of modern breeds

Marina Naval-Sanchez[1], Quan Nguyen[1], Sean McWilliam[1], Laercio R. Porto-Neto[1], Ross Tellam[1], Tony Vuocolo[1], Antonio Reverter[1], Miguel Perez-Enciso [2,3], Rudiger Brauning[4], Shannon Clarke[4], Alan McCulloch[4], Wahid Zamani[5], Saeid Naderi [6], Hamid Reza Rezaei[7], Francois Pompanon [8], Pierre Taberlet[8], Kim C. Worley[9], Richard A. Gibbs[9], Donna M. Muzny[9], Shalini N. Jhangiani[9], Noelle Cockett[10], Hans Daetwyler[11,12] & James Kijas[1]

Domestication fundamentally reshaped animal morphology, physiology and behaviour, offering the opportunity to investigate the molecular processes driving evolutionary change. Here we assess sheep domestication and artificial selection by comparing genome sequence from 43 modern breeds (*Ovis aries*) and their Asian mouflon ancestor (*O. orientalis*) to identify selection sweeps. Next, we provide a comparative functional annotation of the sheep genome, validated using experimental ChIP-Seq of sheep tissue. Using these annotations, we evaluate the impact of selection and domestication on regulatory sequences and find that sweeps are significantly enriched for protein coding genes, proximal regulatory elements of genes and genome features associated with active transcription. Finally, we find individual sites displaying strong allele frequency divergence are enriched for the same regulatory features. Our data demonstrate that remodelling of gene expression is likely to have been one of the evolutionary forces that drove phenotypic diversification of this common livestock species.

[1] CSIRO Agriculture and Food, 306 Carmody Road, St. Lucia 4067 QLD, Australia. [2] Centre for Research in Agricultural Genomics (CRAG), Bellaterra 08193, Spain. [3] ICREA, Carrer de Lluís Companys 23, Barcelona 08010, Spain. [4] AgResearch Ltd, Invermay Agricultural Centre, Private Bag 50034, Mosgiel 9053 Otago, New Zealand. [5] Department of Environmental Sciences, Faculty of Natural Resources and Marine Sciences, Tarbiat Modares University, Noor, Mazandaran 46414-356, Iran. [6] Natural Resources Faculty, University of Guilan, Guilan 41335-1914, Iran. [7] Environmental Science Department, Gorgan University of Agricultural Sciences and Natural resources, Gorgan 49138-15739, Iran. [8] Laboratoire d'Ecologie Alpine, Universite Grenoble Alpes, Grenoble 38041, France. [9] Human Genome Sequencing Center, Department of Human and Molecular Genetics, Baylor College of Medicine, Houston, TX 77030, USA. [10] Utah State University, 1435 Old Main Hill, Logan, UT 84322-1435-1435, USA. [11] Department of Economic Development, Jobs, Transport and Resources, Bundoora 3083 VIC, Australia. [12] School of Applied Systems Biology, La Trobe University, Bundoora 3083 VIC, Australia. Correspondence and requests for materials should be addressed to J.K. (email: James.Kijas@csiro.au)

The domestication of plants and animals commenced around 10,000 years ago and precipitated enormous societal change by transitioning human kind from hunter-gatherers to agricultural settlers[1,2]. The impact on domesticated animals themselves has also been transformative, with radical phenotypic, morphological and behavioural changes occurring in comparison to their wild counterparts. Elucidating the molecular basis of these changes has the potential to assist the process of animal breeding and illuminate how genotypic variation influences phenotype. Analyses using population scale SNP array data[3–5] and more recently whole-genome sequence data[6–10] have identified regions and genes under selection in domestic species. To extend our understanding beyond key genes, multiple studies have sought to identify categories of sites or genomic features over-represented in sweep regions. These have included evolutionarily conserved elements and proximity to transcription start sites[6,8,10]; however, the approach has been severely hampered by the paucity of detailed functional annotation available for livestock animal species. This is starting to be addressed by the Functional Annotation of Animal Genomes (FAANG) consortium; however, human regulatory information remains by far the most abundant courtesy of the ENCODE[11] and Epigenetics Roadmap initiatives[12]. In both, chromatin state and DNA accessibility assays have been used across a plethora of human cell lines and tissues to comprehensively map regulatory features. This has proven invaluable for investigating the link between genetic variation, gene regulation and complex human diseases[13,14], as well as human adaptive traits[15].

Exploiting the wealth of ENCODE data in non-human species is dependent on the strength of sequence and functional conservation. Encouragingly, sequence conservation of non-coding elements has been successfully used to identify functional regulatory sequences between diverse species[16,17]. Further, recent comparative genomic studies in mammals have shown that despite a significant divergence of Transcription Factor Binding Sites (TFBSs)[18,19] and enhancers[20], there is a substantial core of regulatory elements which can be characterised based on sequence conservation[21–23]. For example, Mouse ENCODE comparative analyses concluded that 44% of promoters and 40% of enhancers in mice that mapped to human retained functional and tissue specific activity[22].

In this study, we investigate the impact of sheep domestication and subsequent artificial selection on genomic variation. First, we analyse genome sequences from a global distribution of 43 domestic sheep breeds and 17 Asiatic mouflon, its wild counterpart. Population diversity analysis identified 1420 sweep regions under positive selection in domestic sheep. Next, we use human genome regulatory information to generate a comparative functional annotation of the sheep genome, before assessing it against histone modification data we collected from sheep adipose tissue. Finally, we assess the strength of overlap between selection signatures and genomic regions implicated in regulation of gene expression. We find strong and significant overlap for multiple components of the proximal gene regulatory machinery, including promoters and chromatin states indicative of active transcription. These results provide both a high-resolution genomic view of positive selection and a first draft functional annotation of the sheep genome. Considered together, our results suggest modification to gene regulatory networks has been an important evolutionary driver of the phenotypic changes that distinguish domestic sheep from their wild ancestors.

## Results

**Genetic variation among domestic and wild sheep.** We sequenced the genome of 67 domestic sheep drawn from 43 phenotypically diverse breeds for comparison with 17 Asiatic Mouflon representing their wild ancestor (Fig. 1a, Supplementary Data 1). Sequencing to an average depth of 11.82× coverage was followed by alignment and variant calling to identify 28.1 million high-quality SNPs. Analysis using the protein coding gene set of the reference assembly OARv3.1[24] revealed <1% of SNP (218,762) were located in exons (Supplementary Table 1), and of these the majority (133,891) were synonymous variants (Supplementary Table 2). Despite sampling 43 geographically dispersed breeds, domestic sheep contained fewer private SNP (Fig. 1b, c, Supplementary Table 3, Supplementary Data 2), and lower nucleotide diversity ($\pi$) than wild sheep (domestic sheep $\pi = 0.16\%$ per nucleotide, mouflon $\pi = 0.20\%$; Fig. 1d). Reduced diversity is an expected consequence of domestication, however in sheep the magnitude of change appears smaller than for some other domesticates[8,25]. This suggests sheep domestication captured a wide sampling of diversity from Asiatic mouflon, consistent with the finding that many modern breeds retain high-effective population size[4] and a diverse representation of mtDNA haplogroups[26]. Three metrics that assess genetic relatedness revealed Mouflon and domestic breeds remain relatively closely related. We found 70% of SNP in domestic sheep were also polymorphic in Mouflon (14.3 out of 20.4 M SNP, Fig. 1b) and the allele frequencies of SNP were highly correlated (Fig. 1c). Finally, population differentiation between domestic and wild sheep was low for an inter-species comparison ($F_{ST} = 0.093$). We next evaluated the relationship between individual genomes using Principal Component Analysis (PCA), identifying two distinct clusters of Mouflon from geographically separate areas of their range (Supplementary Fig. 1). PCA of domestic sheep genomes clustered animals according to their geographic origin of breed formation (Fig. 1e). PC1 captured an east–west cline, PC2 a north–south cline and breeds from the Middle East (the domestication centre) took a central position with near zero PC values, recapitulating a phylo-geographic structure consistent with earlier SNP array based investigations[4,27].

**Genomic regions under selection in domestic sheep.** We sought to identify genomic regions impacted by positive selection during the domestication of sheep and/or the artificial selection that followed. By sequencing a globally representative collection of phenotypically diverse breeds and searching for common sweep regions, we expect to enrich for selection events acting during domestication or prior to the global radiation of breeds. However caution is required, as sweep events common to divergent breeds may be the result of much more recent selection[28]. Domestic sheep genomes were interrogated for shared regions characterised by both low diversity and high divergence compared to wild sheep. First, we estimated allele frequency divergence as $F_{ST}$ between species in 20 kb windows (Supplementary Fig. 2a). To assess the direction of selection effect, nucleotide diversity ($\pi$) was estimated separately within domestic and wild sheep, before the ratio of mouflon/domestic $\pi$ was plotted for each genomic bin[9,29] (Supplementary Fig. 2b). Joint evaluation of both metrics defined 1420 bins putatively under selection in domestic animals and a further 28 in Mouflon (P-adj <0.01, Fig. 2a, Supplementary Data 3–4). Physically co-located bins (within 50 kb) were joined to define 635 regions (average size 36.3 kb) containing 430 genes (Supplementary Data 5–6). Inspection revealed many genes have previously been shown to be influenced by selection in livestock or associated in sheep biology (Table 1, Supplementary Table 4). These include *SOCS2* for animal weight and milk production[30], *ITCH–ASIP* for coat colour[31] and *VEGFA* involved in early development and reproductive success[4,32]. Additional genes implicated in other domesticates, include *SOX2* in rabbits[8],

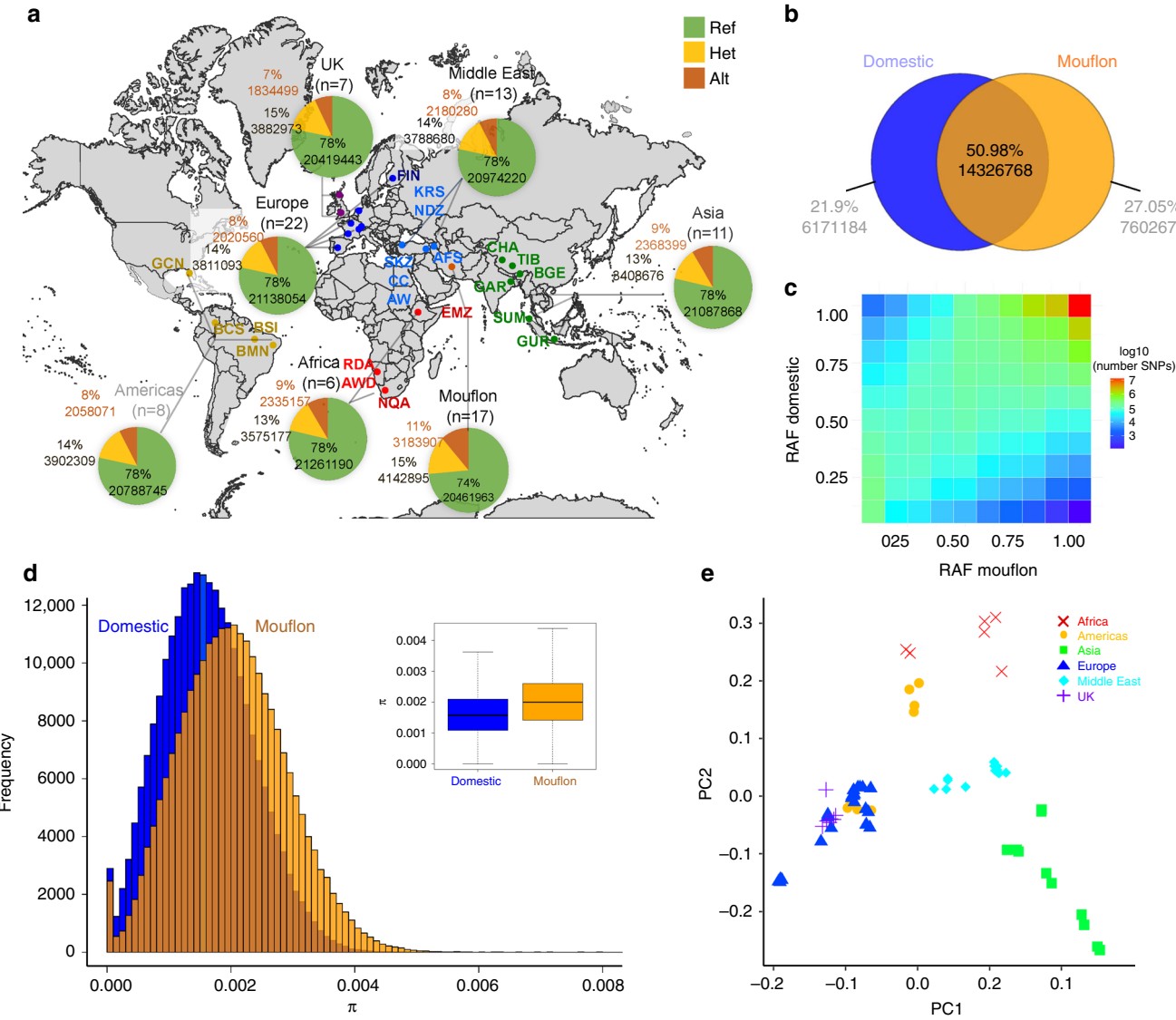

**Fig. 1** Genome diversity and relatedness of wild and domestic sheep. **a** The geographic distribution of 43 breeds sampled for whole-genome sequencing. Population based proportion of allele type is given for each major geographic group of breeds, after calling reference variants present in the Texel derived genome assembly OARv3.1. **b** Proportion and number of private and shared SNP for the collection of wild and domestic sheep genomes. **c** Reference allele frequency (RAF) correlation between domestic and mouflon populations. RAF domestic and mouflon represent the frequency of the reference allele in domestic and mouflon populations respectively. (Pearson correlation: 76.36% *p*-value <2.2E−16). **d** Distribution of nucleotide diversity ($\pi$) within species, estimated within 20 kb bins. The boxplot compares the bin mean and variance. The distribution means, evaluated using the Wilcoxon rank-sum test, are significantly difference (*p*-value <2.2E−16). **e** Principal component (PC) analysis clustering individual domestic sheep coloured to reflect the geographic origin of breed development

*NR6A1* associated with vertebrae number in domestic pig[7], *TBX3* associated with pigmentation in horses[33] and *ESR1* associated with prolificacy in pigs[34] (Supplementary Fig. 3a). One prominent example is *KITLG*, a protein involved in melanocyte development and pigmentation in a variety of mammalian species[35–39]. In sheep, selection sweeps flank either side of the *KITLG* coding exons suggesting they flag alleles with a gene regulatory function (Fig. 2b, c). Another example is the *NCAPG–LCORL* locus implicated in controlling height and stature, however it is unclear which gene exerts the effect[7,40,41]. Domestic sheep contain a selective sweep spanning the *LCORL* promoter region, while mouflon contain a sweep downstream of *NCAPG* (Supplementary Fig. 3b). Many of these candidate genes are likely to contribute to phenotypic variation, however in the absence of individually recorded trait data analysed using an approach, such as GWAS, establishing the direct link between these genes and

their functional consequence is difficult using selection sweep methodology alone.

**Biological processes affected by domestication and selection.** To identify the biological processes influences by domestication and artificial selection, we mapped the ovine sweep regions to the human genome and examined their ontological enrichment using GREAT[42] (see Methods section and Supplementary Data 7). This revealed a significant enrichment (P.adj <0.05 hypergeometric and binomial test) for processes consistent with the evolution of domestic animals from their wild ancestors. One prominent example is dorsal/ventral patterning (Table 2, Supplementary Data 7) as reduction in body size was a quick and early response to domestication across species[43]. A second is regulation of lipid metabolism which may reflect human mediated alteration of

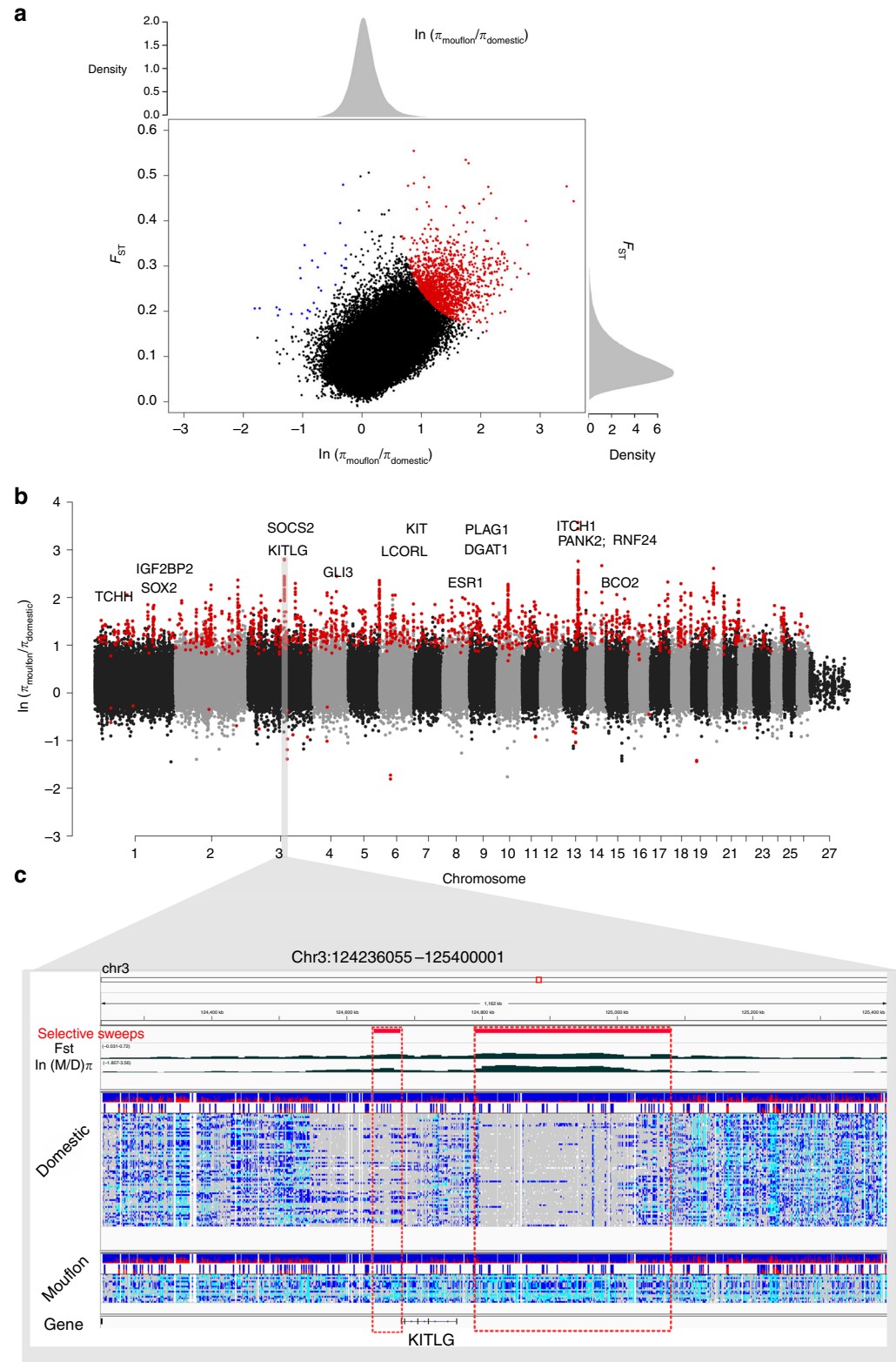

**Fig. 2** Genomic regions putatively under positive selection in sheep. **a** Population differentiation ($F_{ST}$) and relative nucleotide diversity between wild and domestic sheep ($\pi_{mouflon}/\pi_{domestic}$) was estimated in 20 kb genomic bins. A total of 1420 outlier bins exhibiting evidence for selection in domestic sheep genomes are indicated in red (corresponding to Z-test $P < 0.001$, where $F_{ST} > 0.156$ and ln ratio $>0.672$). **b** Genome-wide distribution of relative nucleotide diversity. Positive values identify genomic bins with depressed diversity in domestic sheep compared with mouflon, consistent with positive selection sweeps involved with domestication and selection (Table 1, Supplementary Table 4). **c** Selective sweeps located either side of *KITLG* coding exons. Integrated Genome Visualisation (IGV) screenshot of chromosome 3:124236035–125400001 illustrates the reduction of SNP variation in 67 domestic genomes compared with 17 Mouflon. The two regions identified by our selection metrics are shown inside the dashed red boxes

**Table 1 Genes associated with selective sweeps in domestic animals reported in other studies**

| Chr | Start | End | Avg rnk $F_{ST}$ and $\pi$ | $\pi$ | $F_{ST}$ | Closest genes | Distance to closest gene | Function |
|---|---|---|---|---|---|---|---|---|
| 13 | 50310001 | 50650000 | 2 | 3.56 | 0.53 | *PANK2* | 0 | Neurodegeneration |
| 3 | 124790001 | 125080000 | 21.5 | 2.80 | 0.35 | *KITLG* | 27570 | Coat colour |
| 2 | 184990001 | 185040000 | 42.5 | 1.98 | 0.33 | *GLI2* | 139099 | Growth |
| 1 | 199640001 | 199790000 | 51 | 1.79 | 0.35 | *IGF2BP2* | 0 | Adiposity |
| 4 | 78890001 | 79000000 | 51.5 | 2.44 | 0.31 | *GLI3* | 29289 | Pigmentation |
| 17 | 59450001 | 59520000 | 62.5 | 2.32 | 0.30 | *TBX3* | 215541 | Pigmentation |
| 9 | 36140001 | 36240000 | 63 | 1.72 | 0.34 | *PLAG1* | 0 | Fertility, stature |
| 15 | 21900001 | 22010000 | 70.5 | 1.60 | 0.39 | *BCO2* | 0 | Yellow-fat |
| 9 | 13570001 | 13610000 | 82.5 | 1.85 | 0.30 | *DGAT1* | 0 | Milk fat |
| 6 | 37420001 | 37510000 | 144 | 1.56 | 0.29 | *LCORL* | 0 | Weight/height |
| 3 | 124640001 | 124680000 | 184 | 1.69 | 0.26 | *KITLG* | 0 | Coat colour |
| 8 | 75670001 | 75700000 | 228.5 | 1.97 | 0.24 | *ESR1* | 0 | Litter size; prolifacy |
| 6 | 38500001 | 38530000 | 245.5 | 1.62 | 0.25 | *LCORL* | 1047670 | Weight/height |
| 5 | 19460001 | 19480000 | 312.5 | 1.23 | 0.27 | *IRF1* | 5040 | Immune function |
| 3 | 129730001 | 129750000 | 359 | 0.94 | 0.31 | *SOCS2* | 7494 | Weight and milk production |
| 1 | 100990001 | 101010000 | 383.5 | 1.26 | 0.24 | *TCHH* | 0 | Hair |
| 4 | 78800001 | 78820000 | 386.5 | 1.15 | 0.26 | *GLI3* | 209289 | Pigmentation |
| 5 | 85880001 | 85900000 | 405 | 1.11 | 0.26 | *MEF2C* | 173852 | Skeletal muscle development |
| 2 | 105830001 | 105850000 | 417.5 | 1.27 | 0.23 | *HAND2* | 235173 | Limb development |
| 20 | 17530001 | 17550000 | 434.5 | 1.207 | 0.24 | *VEGFA* | 147890 | Reproduction |
| 13 | 63380001 | 63400000 | 469 | 1.237 | 0.22 | *ITCH* | 0 | Coat colour |
| 1 | 203460001 | 203480000 | 481 | 0.977 | 0.25 | *SOX2* | 52282 | Stem cell maintenance |
| 6 | 35290001 | 35310000 | 486.5 | 1.05 | 0.24 | *GPRIN3* | 201294 | Brain development |
| 6 | 70290001 | 70310000 | 492.5 | 1.24 | 0.21 | *KIT* | 55390 | Coat colour |

**Table 2 Gene ontology biological process—top 15 enrichment results for identified selective sweeps (Great v.3.0)[42]**

**GO biological process**

| Term name | Binom raw *P*-value | Binom FDR Q-Val | Binom fold enrichment | Binom observed region hits | Binom region set coverage |
|---|---|---|---|---|---|
| Primary alcohol catabolic process | 1.12E−27 | 2.34E−24 | 23.4 | 27 | 0.020 |
| Regionalisation | 3.22E−19 | 9.62E−17 | 2.3 | 135 | 0.103 |
| Regulation of MAPK cascade | 5.84E−19 | 1.60E−16 | 2.1 | 173 | 0.132 |
| Development of primary male sexual characteristics | 6.82E−19 | 1.83E−16 | 3.1 | 83 | 0.063 |
| Gland development | 8.90E−19 | 2.21E−16 | 2.3 | 138 | 0.105 |
| Male gonad development | 2.34E−18 | 5.20E−16 | 3.3 | 73 | 0.055 |
| Male sex differentiation | 4.05E−18 | 7.98E−16 | 3.0 | 83 | 0.063 |
| Dorsal/ventral pattern formation | 9.33E−18 | 1.68E−15 | 3.5 | 67 | 0.051 |
| Regulation of lipid metabolic process | 2.58E−17 | 4.09E−15 | 2.7 | 92 | 0.070 |
| Negative regulation of cell cycle | 6.13E−17 | 9.28E−15 | 2.6 | 97 | 0.074 |
| Embryonic hemopoiesis | 1.30E−16 | 1.80E−14 | 6.0 | 35 | 0.027 |
| Odontogenesis | 8.84E−16 | 1.04E−13 | 3.2 | 66 | 0.050 |
| Branching morphogenesis of an epithelial tube | 1.43E−15 | 1.61E−13 | 2.63 | 91 | 0.069 |
| Negative regulation of cellular protein metabolic process | 2.00E−14 | 1.65E−12 | 2.1 | 127 | 0.097 |
| Development of primary sexual characteristics | 3.83E−14 | 2.96E−12 | 2.2 | 105 | 0.079 |

muscularity and fatness as sheep were actively managed as a food source[44]. A third example involves sexual maturation, as humans increasingly exerted control over breeding, pressure to maintain beneficial sexual fitness traits in the wild were removed, reducing sexual dimorphism and altering the timing of reproduction[45]. The residual consequences of these changes were identified in the data, as we identified significant biological processes associated with sexual differentiation, particularly in male sexual characteristics and differentiation (Table 2, Supplementary Data 7). We next evaluated sheep sweep regions against known mouse mutants and their phenotypic consequences. This identified strong enrichment for traits, including oogenesis, litter size, sexual maturation and ovary morphology (Supplementary Data 8).

**Preliminary functional annotation of the sheep genome**. In the absence of detailed functional annotation for any livestock genome, we used comparative genomics to predict ovine regulatory elements using human data[11,12]. Reciprocal liftOver was used to predict the ovine genome location of ENCODE promoters and enhancers, along with 12 chromatin states built using 127 diverse epigenomes[12] (Fig. 3a, see Methods section). ENCODE histone modification marks and transcription factor binding sites for proximal promoters and distal enhancers were predicted with similar mapping efficiency (Supplementary Table 5) while higher variability was observed across the set of 12 chromatin states. Approximately 40–50% of active state elements, or elements associated with expressed genes, were reciprocally mapped

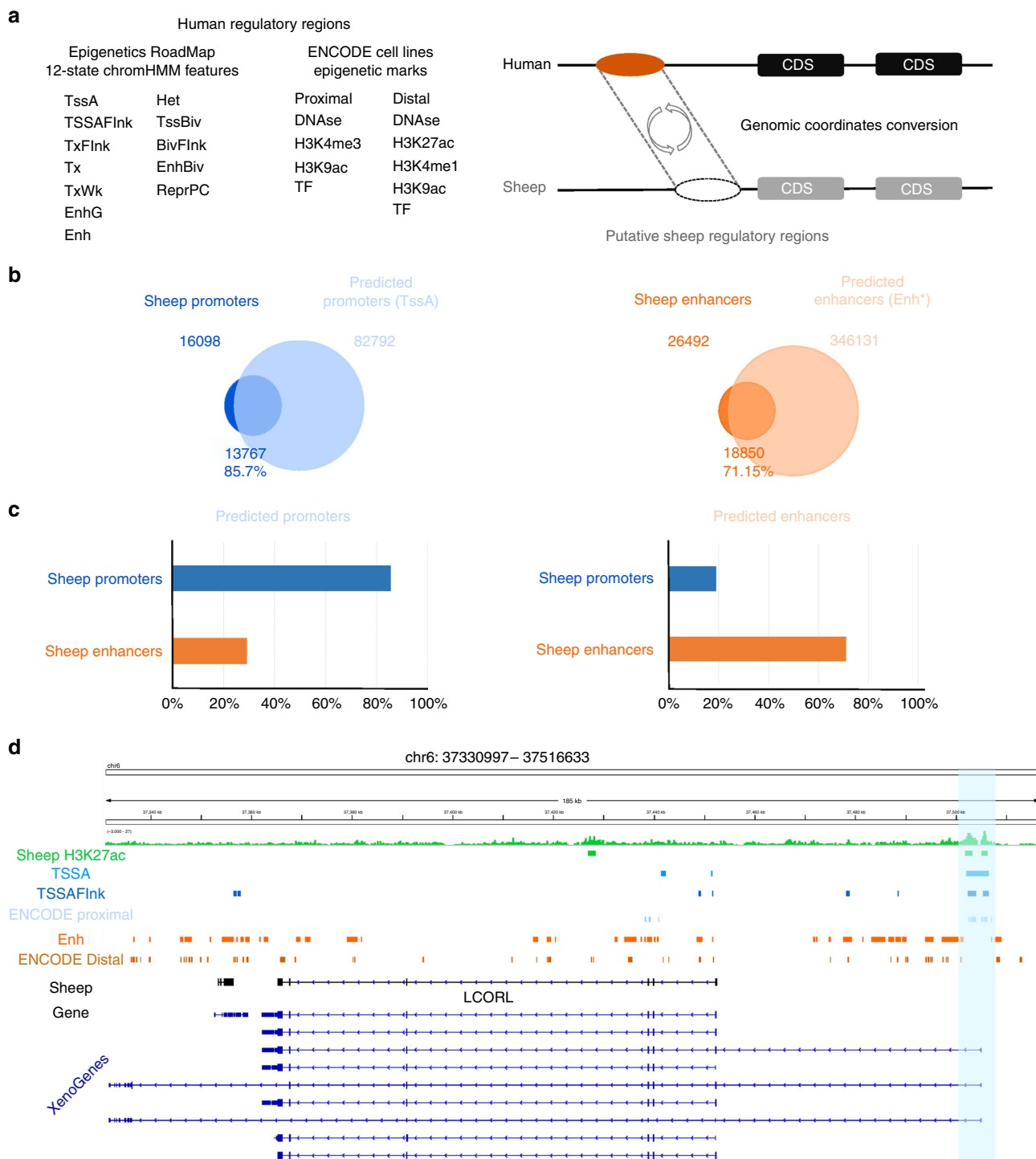

**Fig. 3** Prediction and validation of sheep gene regulatory elements. **a** Human Epigenome Roadmap chromatin states and ENCODE features used in a reciprocal liftOver protocol to predict regulatory regions in sheep. **b** Overlap between chromosomal locations identified by experimental sheep tissue ChIP-Seq (H3K4me3 and H3K27ac) and regions predicted by liftOver. H3K4me3 peaks are compared against predicted Roadmap chromatin state TssA identifying active promoters (left) and unique H3K27ac peaks are compared to the Roadmap Enh state, excluding those regions overlapping with predicted TssAs, identifying distal enhancers (right). **c** The proportion of predicted promoters (left) and enhancers (right) recovered by ChIP-Seq using either H3K4me3 peaks (blue bar) or unique H3K27ac peaks (orange bar). This demonstrates the predicted annotations retain a level of specificity between components of the gene regulatory apparatus. **d** An example annotated region containing *LCORL* highlights the correspondence between ChIP-Seq data (Sheep H3K27ac) and liftOver predictions (TSSA, TSSAFlnk, ENCODE proximal, Enh, ENCODE distal)

including active TSS promoter states (TssA 41%; TssAFlank 48.5%), and transcription at the 5′ and 3′ ends of genes (TxFlnk 40%). Mapping efficiency broadly reflected evolutionary conservation for each genomic feature as measured by enrichment for mammalian conserved elements[12].

**Human epigenetic data captures sheep regulatory elements.** To evaluate if our comparative genomics approach successfully predicted regulatory elements in sheep, we performed H3K4me3 and H3K27ac ChIP-seq analysis of sheep late gestation perirenal brown adipose tissue. H3K4me3 identifies active gene promoters[46,47], and we identified a total of 16,098 regions. Comparing their genomic location against Roadmap active promoter state TssA predictions revealed an 85% recovery rate (Fig. 3b, 13,767/16,098). A large number of predicted promotors were not present, possibly reflecting our ChIP-Seq data originate from only one sheep tissue and timepoint. H3K27ac identifies both active promoters and active enhancers, and 35,366 regions were identified. Of these, 26,492 regions were uniquely identified by H3K27ac but not H3K4me3 and defined in this study as enhancers. Comparing the location of these 26,492 experimentally defined enhancers with predicted enhancer state Enh, excluding those overlapping the defined state TssA, revealed a recovery rate of 71% (Fig. 3b, 18,850/26,492). The recovery rate and significance for each Roadmap and ENCODE predicted genomic feature is given in Supplementary Fig. 4–7 and Supplementary Table 6. This revealed the predicted features were significantly enriched for sheep regulatory elements and retained specificity between different components of the gene regulatory machinery. For example, H3K4me3 marks recovered 85% of predicted promoters but only 29% of enhancers while H3K27ac unique marks recovered only 19% of predicted promoters but 71% of enhancers (Fig. 3c). Next, we performed H3K27me3 ChIP-Seq in sheep brown adipose tissue, a mark related to Polycomb repression. This yielded 31,942 elements. Comparison against the set of predicted regulatory features revealed strong enrichment only for repressive features, including repressed Polycomb and bivalent/poised TSSs (Supplementary Table 6). This provides confidence our comparative approach successfully recovered biologically meaningful genome feature annotations with the sensitivity to discriminate promoters, enhancers, and repressive regulatory elements. An example of a predicted functional annotation is given in Fig. 3d. Finally, we sought to explore how robust the reciprocal liftOver approach is to the combination of species in which it is applied. Similar results were obtained when using human Roadmap elements to predict promoters and enhancers in the mouse, cow and pig genomes, where in each case available histone modification mark data sets were strongly enriched for predicted elements (Supplementary Fig. 8, Supplementary Table 7).

Three additional properties of the predicted genome features were assessed to evaluate their quality. First, the distance from each feature to the nearest transcription start site (TSS) was assessed for both the sheep and human genome. This revealed that ENCODE predicted proximal promoter elements were consistently closer to TSS in the sheep genome than distal enhancers (Supplementary Fig. 9). Second, levels of nucleotide diversity within each predicted annotation feature were compared. This showed epigenomic modifications reporting proximal promoters contained lower diversity than those marking distal enhancers, consistent with findings for other species[48] (Supplementary Fig. 10). Third, nucleotide diversity was compared between elements active in either a ubiquitous or tissue specific manner. Predicted promoters active in many Roadmap epigenomes (>100 cell types; TssA $\pi < 0.06$; Enh $\pi < 0.15$) had higher evolutionary constraint compared with promoters expressed in a restricted

manner (<10 cell types, TssA $\pi < 0.09$; Enh $\pi < 0.12$, Supplementary Fig. 11). Taken together, the results from our sheep specific histone modification data and the analysis of nucleotide diversity validate that the collection of predicted annotated features are of sufficient utility to address evolutionary questions.

**Selection sweeps are enriched for proximal features.** A key objective of the study sought to investigate if ovine genomic regions putatively affected during domestication and artificial selection displayed an enrichment of functional domains associated with protein translation, transcription or the regulation of gene expression. As input we used the collection of genome features described above that included: (i) sheep H3K4me3 and H3K27ac ChIP-Seq peaks; (ii) Epigenome Roadmap predicted chromatin states; (iii) ENCODE predicted proximal and distal elements, and; (iv) gene model annotations from the reference assembly OARv3.1. To test for enrichment, a location overlap approach developed for interpretation of epigenomic data (LOLA)[49] was used to assess each genomic feature against the distribution of sweep regions (1420 bins, Fig. 2a). This revealed selection sweeps strongly co-localised with gene regulatory elements after correction for multiple testing (Fig. 4a). Three components of the proximal gene regulatory machinery were significant: enhancer elements physically associated with genes (EnhG, LogP = 12.83), actively transcribed states (Tx, LogP = 10.68) and proximal promoter states (TxFlnk, LogP = 8.03); (Fig. 4a, Supplementary Table 8–9). Next, we repeated the analysis after linking neighbouring outlier bins (within 50 kb) to define 635 selective sweep regions. This was a more permissive analysis as the genome fraction implicated with domestication and artificial selection was higher (Methods section). Sweep regions were significantly enriched for 14 genome features (Supplementary Fig. 12, Supplementary Table 10). Strikingly, the eight most strongly enriched were each components of the proximal gene regulatory machinery and active transcription including sheep ChIP-Seq modifications, Epigenome Roadmap states (TxFlnk, EnhG, Tx, TssA), proximal ENCODE features obtained by DNase I signal and transcription factor binding sites and translated exons (CDS, Supplementary Fig. 12).

**Site frequency analysis.** Domestication and selection are anticipated to have altered the allele frequency of loci mechanistically involved in the phenotypic and molecular changes that distinguish domesticates from their wild ancestors. We therefore applied site frequency analysis to identify nucleotides with divergent allele frequency when compared between wild and domestic sheep genomes. This exploits the availability of whole-genome sequence by assessing every site, and is independent from the selection sweep methodology used above. We first estimated allele frequencies within domestic and wild sheep using 14 M sites found to be segregating in both species. Comparison between species revealed the majority displayed highly correlated allele frequencies, with 8.9 M sites (63% of 14 M, Supplementary Data 9) having allele frequency difference (ΔAF) <0.2. This is consistent with the behaviour of neutral loci between closely related species. Only a very small fraction of loci exhibited strongly divergent allele frequency. These are characteristic of sites under selection, however they can be difficult to distinguish from those impacted by genetic drift alone. A total of 9161 SNP (0.016%) had ΔAF >0.8 and only 1059 sites had divergence approaching the maximum (>0.9, Supplementary Data 10). We expect this set of loci are likely to have played an outsized role in domestication and selection, prompting us to characterise them in more detail. To evaluate the impact of gene inactivation we searched these sites for nonsense mutations introducing stop

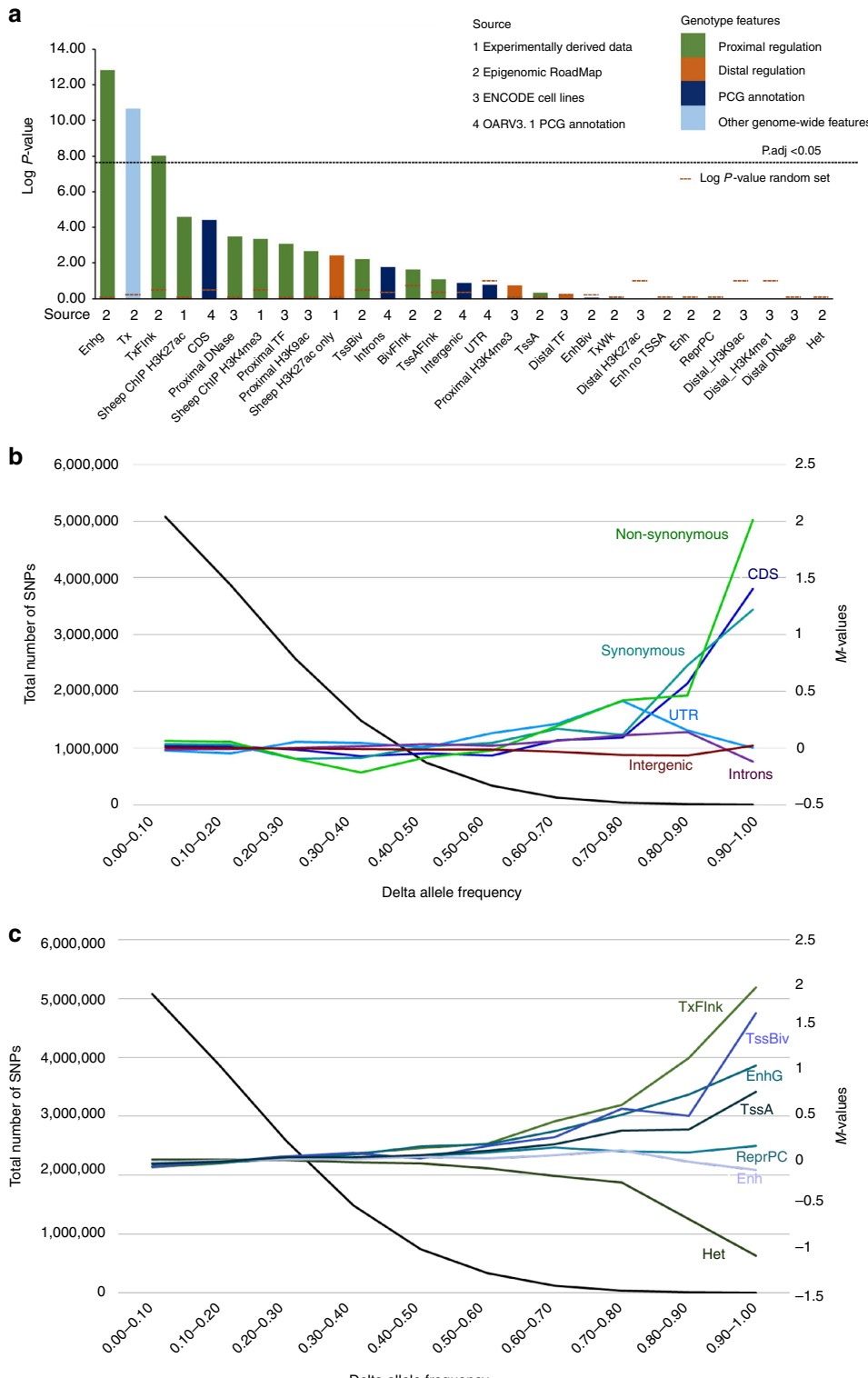

**Fig. 4** Genomic feature enrichment in selection sweeps and differentiated sites. **a** Strength of enrichment for 29 genome features within 1420 sweep bins assessed by location overlap[49]. Genome features were derived from four different sources. The significance threshold from multiple testing is represented by the horizontal line. **b** Intersection of delta allele frequency (ΔAF) with protein coding gene annotations from reference OARv3.1. The number of SNP in ΔAF bins is given at left, and the *M*-value (at right) was calculated by comparing the frequency of SNP in each genome feature and ΔAF bin with the corresponding frequency across all bins. **c** As for **b** using chromatin state annotations derived from the Roadmap dataset. Additional *M*-value results using predicted ENCODE marks are provided in Supplementary Fig. 13

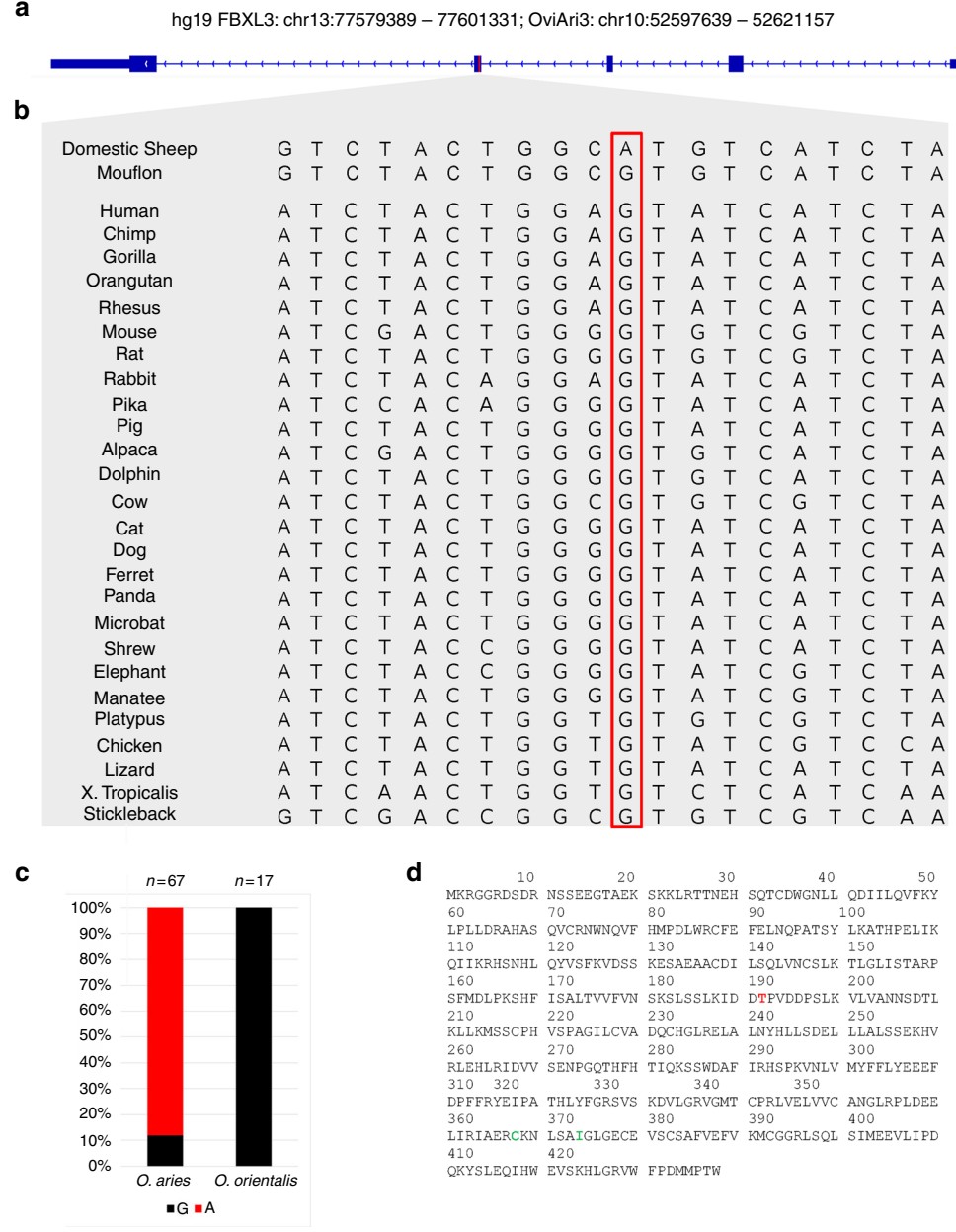

**Fig. 5** Candidate causal missense mutation in *FBXL3*. **a** *FBXL3* gene representation in human hg19 coordinates chr13:77579389-77601337. **b** Multiple sequence alignment of SNP across representative vertebrates. **c** Allele frequency of the G (reference) or A (alternative) allele in domestic sheep (*O. aries*) (*n* = 67) or mouflon (*O. orientalis*) (*n* = 17). **d** *FBXL3* amino acid sequence with the substitution at residue 182 highlighted in red. Residue 358 (green) is associated with the mouse After hour (Afh) phenotype residue 364 with the overtime phenotype (Ovt)

codons or that modify the splicing machinery. This revealed a single variant, located in a splice donor site of *MICAL3*, was present in sites with extreme ΔAF (>0.8) (Supplementary Data 9). Therefore, as in chicken[6], rabbit[8] and pigs[7], our conclusion is that gene inactivation via nonsense (frame-changing) and splice site mutation has not played a major role during sheep domestication. We then performed enrichment analysis to evaluate the number of SNP across ΔAF bins compared to their location within annotated genome features. This approach has been used previously[8,50] and captures data from every segregating SNP available. Assessment of protein coding gene annotations derived from OARv3.1 revealed non-synonymous substitutions were strongly enriched for SNP in high-ΔAF bins, while no relationship was detected for introns, UTR or intergenic regions (Fig. 4b). Regulatory elements are embedded within the intergenic

genome feature; however, they likely represent an insufficiently large proportion to cause intergenic regions to display enrichment. Our result do demonstrate that selection within coding regions has clearly played a role in adaptive change during domestication and selection. Of 27 SNP with ΔAF>0.80, there is only one missense mutation where domestic sheep carry an allele different from other vertebrate genomes (Fig. 5). The variant is in *FBXL3* (F-Box And Leucine-Rich Repeat Protein 3) which is known to be involved in regulation of circadian rhythms[51,52] and mice with missense mutations have an altered circadian period[53,54]. Many modern sheep breeds have elongated reproductive seasonality compared to wild sheep, and seasonality is photoperiod dependent. It is therefore possible the *FBXL3* mutation identified here plays a role in controlling seasonality in sheep.

In addition to analysis of protein coding gene annotation, the same site frequency approach was applied to the RoadMap and ENCODE predicted features (Fig. 4c, Supplementary Fig. 13). Striking enrichment was detected for chromatin states associated with both active and developmentally poised promoters (TxFlnk, TssBiv), as well as proximal enhancers (EnhG, Fig. 4c). Further, analysis of ENCODE derived features show enrichment for proximal elements that was absent for each distal category of regulatory element (Supplementary Fig. 13). We artificially altered exonic allele frequencies to search for any impact on other annotation features, and found modest enrichment that serves as a note of caution (Supplementary Fig. 14). None the less, the findings are remarkably similar to the feature enrichments present in the selective sweep analysis (Fig. 4a), thus providing evidence that domestication and artificial selection have preferentially impacted the proximal gene regulatory machinery.

## Discussion

We explore the genomic consequence of domestication and subsequent artificial selection by sequencing 43 breeds of domestic sheep and comparing them to descendants of their wild ancestors. The data revealed both protein coding genes and proximal regulatory elements played major roles during the evolution of modern sheep, addressing a long standing question in evolutionary biology concerning the relative contribution played by regulatory versus protein coding sequence on phenotypic variation[55,56].

Genome sequencing clearly showed protein coding genes have been directly altered during the development of modern sheep. Genomic regions with evidence of positive selection were significantly enriched for coding exons, and missense (non-synonymous) substitutions were fourfold over represented in sites with extreme allele frequency differences between wild and domestic genomes. Sweep regions contained a collection of genes previously implicated in the control of pigmentation, reproduction and stature and biological process enrichment was observed for sexual differentiation and altered timing of reproduction.

A more difficult task has been any detailed evaluation of the contribution made by sequence elements that regulate gene expression. To address this we built a comparative functional annotation of sheep and validated its utility by comparison with experimentally derived histone modification data identifying active sheep promoters, enhancers and repressors. Our major finding is multiple chromatin states and regulatory elements are strongly enriched in both selective sweeps and sites with divergent allele frequency. Involvement of the gene regulatory machinery is consistent with the emerging view that complex trait architecture is often governed by changes in gene expression. For example, the majority of trait associated SNP in human disease GWAS fall outside transcribed regions[57]. Intersecting these associated SNP with epigenomic marks reveals strong enrichment within both ENCODE[57] and Roadmap genome features[12]. Similarly in livestock, testing production trait associated SNP has revealed enrichment within non-coding elements neighbouring genes, suggesting a role for variants that influence gene expression[58]. Next, evo-devo studies have put forward the relevance of regulatory mutations affecting morphology, behaviour and adaptation[59,60]. Seminal examples include *Drosophila* wing pigment patterns[61] and skeletal reduction in stickleback fish[62]. Perhaps of most relevance to animal domestication, analysis of modern rabbits suggests a strong enrichment has occurred at conserved non-coding sites expected to be enriched for regulatory function[8]. In addition, the earliest stages of trout domestication are characterised by large, heritable changes to gene expression[63]. Our analysis, armed with the extensive epigenomic annotation described here, finds evidence that remodelled gene regulation has been a force during the profound morphological, behavioural and metabolic changes occurring through the transition to a domesticated state.

The richness of predicted epigenomic feature annotations allowed us to discriminate between the components of the gene regulatory apparatus that have been preferentially impacted. Perhaps surprisingly, active promoters and proximal enhancers were strongly enriched in both sweep regions and sites with divergent ΔAF, in contrast to each distal ENCODE mark or inactive Roadmap chromatin states, such as repressed Polycomb and constitutive heterochromatin. This suggests remodelling of control elements immediately adjacent to genes has been a dominant mechanism of change, however it is currently unknown why distal elements appear to have made a diminished contribution.

Most experiments into the consequences of animal domestication and selection have relied on SNP array data and have focussed on protein coding genes in the absence of functional annotation[3–5]. Genome sequencing has dramatically improved the resolution of selective sweep regions in this study of sheep and other studies of pigs and chickens[6,64,65]. Further, our comparative application of human epigenomic data has proven highly insightful, however limitations remain. The reciprocal BLAST methodology may have introduced bias, resulting in an over-representation of evolutionarily conserved promoter elements compared with enhancers. Indeed the recovery of enhancers within sheep ChIP-Seq data was lower than for promoters, however the difference was not large and enhancers have a higher evolutionary turnover compared to proximal regulatory elements or protein-coding sequences[48,62,66]. Further, it is possible that species specific and tissue specific regulatory elements are likely to be under represented as a consequence of both our methodology and the application of ChIP-Seq using only a single tissue. A separate note of caution relates to the sampling of animals sequenced in the study, which is likely to have an impact on the final collection of sweeps identified as the choice of genomes used has a measurable impact on allele frequencies (Supplementary Fig. 15). A final challenge arises from linkage disequilibrium and the overlapping nature of some genome annotation features, which together blur the relative contribution of physically co-located genomic features (Supplementary Fig. 14). A case in point is the observation that synonymous sites displayed enrichment in our site frequency analysis. It is not clear if this reflects a functional role for many synonymous positions via cryptic effects on transcription or mRNA transport as previously suggested[67] or if their proximity to missense mutations has generated the enrichment via linkage disequilibrium. Despite these limitations, the collection of annotated genome features represents a useful resource for the research community to interpret GWAS and inform the design of SNP panels to increase power to deliver genomic prediction. Knowledge concerning the elements likely to regulate gene expression enhances our understanding of the sheep genome, and suggests that remodelling gene expression has been a key mechanism during the evolution of this important livestock species.

## Methods

**Samples.** A total of 70 animals were sampled from 43 domestic breeds and subjected to genome sequencing (Supplementary Data 1). These comprise 46 animals selected from an earlier SNP array based global survey of breed diversity[4] and another six animals used for SNP discovery, construction of the SNP50 BeadChip and CNV detection. The final group of 18 individuals have not been examined before. Breeds were drawn from Asia (12), Africa (6), the Middle East (13), the Americas (8), the United Kingdom (8) and continental Europe (23). Whole-genome sequence data for 19 Asian mouflon (*Ovis orientalis*) was collected and made available by the NEXTGEN project (http://nextgen.epfl.ch/). Fastq files were

downloaded from the ENA public repository (http://www.ebi.ac.uk/ena/data/view/PRJEB3139) and processed as described below for the domestic sheep genomes.

**Genome sequencing, variant detection and annotation.** Paired-end short insert libraries were constructed using 5 μg of genomic DNA and sequenced on the Illumina HiSeq 2000 platform. Reads were mapped against the sheep reference assembly v3.1[24] using BWA aligner v0.7.12 (bwa aln + bwa sampe, default parameters)[68]. Animals were sequenced to an average median depth of 11.8 × (8.4–17.2×) (Supplementary Data 1). Duplicate reads were removed using Picard tools (http://broadinstitute.github.io/picard/), and local realignment around INDELS was performed using GATK v3.2.0[69]. Variant detection and SNP diversity analyses were performed using SAMTOOLS 1.2.1 mpileup and annotated using VCFtools v0.1.14[70]. After obtaining genotype calls for a total of 89 samples the following filters were applied using a combination of VCFtools[70] and in-house scripts: (i) SNP were retained in positions with read depth between 5× and twice the average depth per sample; (ii) minimum mapping quality of 30 and base quality of 20 were applied; (iii) SNP within 5 bp of INDELS were removed; (iv) for SNP pairs separated by <4 bp, the lower quality variant was excluded; (v) tri-allelic variants were removed; (vi) SNP called in <90% of animals were excluded and (vii) SNP displaying an excess of heterozygosity were excluded (--hwe 0.001). This defined a set of 28,100,631 SNP across domestic (67) and mouflon (17) genomes. A total of five low-coverage animals were excluded (3 domestic and 2 mouflon). PLINK v1.9 was used to perform genetic diversity estimates and PCA (https://www.cog-genomics.org/plink2)[71]. The variant effect predictor tool from ensembl (version 78) was used to identify 24 separate SNP classifications, including coding, missense and non-synonymous substitutions, intron and intergenic, in relation to the gene models annotated on reference assembly OARv3.1[24] (Supplementary Tables 1–2).

**Genome scanning for selection sweeps.** Population differentiation between wild and domestic populations were measured as $F_{ST}$ using the Weir and Cockerham method[72]. Average SNP $F_{ST}$ values were plotted in 20 kb genomic bins with a 10 kb step (Fig. 2a; Supplementary Fig. 2). Nucleotide diversity ($\pi$) was estimated for the same bins in both wild and domestic sheep genome collections as the number of heterozygous SNP by bin size. Genomic bins with fewer than 20 SNP were excluded. Positive selection is characterised by reduced $\pi$, prompting us to estimate a ratio (wild/domestic) that identifies the direction of selection[9,29]. The ratio (wild/domestic) is given in Fig. 2a and plotted in genomic order (Supplementary Fig. 2). The genome-wide threshold to declare significance of Z-transformed values ($F_{ST} > 0.156$ and log $\pi$ ratio >0.672) represents Bonferroni adjusted p-value of <0.01.

**Biological process enrichment within selective sweep regions.** Locus-based Gene Ontology enrichment was performed using GREAT v3.0.0[42]. Genomic bins with evidence of positive selection were translated to human coordinates (GRC37/hg19) using UCSC's liftOver tool (minMatch = 0.1)[73]. The membership and frequency of gene regulatory domains present was then compared against a background set representing all genomic bins using two statistical tests. A binomial and a hypergeometric test assessed the enrichment of molecular function terms and biological process terms. We also used GREAT to interrogate the Mouse Genome Informatics (MGI) database phenotypes[74].

**Building a comparative sheep functional annotation.** Our approach exploited the wealth of functional annotation data generated by the Epigenome Roadmap and ENCODE studies[11,12]. We performed reciprocal liftOver (minMatch = 0.1)[73], meaning elements that mapped to sheep also needed to map in the reverse direction back to human with high quality. This bidirectional comparative mapping approach was applied to 12 chromatin states defined using 5 core histone modification marks, H3K4me3, H3K4me1, H3K36me3, H3K9me3 and H3K27me3. Mapping success is given in Supplementary Table 5. The same approach was applied to ENCODE marks derived from 94 cell types (https://www.encodeproject.org/data/annotations/v2/) with DNase-seq and TF ChIP-seq.

**ChIP-seq of ovine adipose tissue.** Chromatin immunoprecipitation followed by next generation sequencing (ChIP-Seq) of the histone chromatin modification H3K4me3, H3K27ac and H3K27me3 was undertaken using late gestation fetal perirenal adipose tissue (PRAT). This tissue was selected as it may be important in a production setting, where it influences lamb survival under some circumstances. All procedures involving animals were carried out with approval from the University of Adelaide Animal Ethics Committee. The adipose tissue was taken at 130 days post conception from three animals for H3K4me3 and two animals for H3K27ac and H3K27me3. Input nucleosomal DNA from the combined fetal samples were used as input controls. Nuclei were isolated from 1.2 g of frozen PRAT by pulverising the tissue under liquid nitrogen at 4°C in Nuclei Buffer (0.3 M sucrose, 60 mM KCl, 15 mM NaCl, 5 mM MgCl$_2$, 0.1 mM EGTA, 15 mM Tris-HCl, pH 7.5) with the inclusion of protease inhibitors (0.5 mM DTT, 0.1 mM AEBSF, CompleteTM EDTA-free protease inhibitor cocktail (Roche Diagnostics GmbH, Mannheim Germany) using a hand held Dounce homogeniser. The filtered homogenate was centrifuged (3000×g, 5 min) and the pellet recovered and re-suspended in ice-cold Nuclei Buffer containing 0.4% IGEPAL CA-630 (Sigma),

incubated on ice for 5 min and the suspension was then layered on a cushion of Nuclei Buffer containing 1.2 M sucrose. The sample was centrifuged at 10,000×g (25 min, 4 °C), the chromatin pellet recovered and resuspended in micrococcal nuclease digestion buffer. The isolated chromatin was treated with micrococcal nuclease (MNase) (0.2 U/μl) (New England BioLabs, Ipswich, MA, USA) and soluble chromatin recovered using mild sonication and centrifugation. Specific antibodies to H3K4me3 (Abcam ab8580) and H3K27ac (Abcam ab4729) and H3K27me3 (Millipore 07-449) (10 μg per incubation) were used for immunoprecipitation (4 °C, gentle mixing, 15 h). Immune complexes were isolated using Protein A-Sepharose (4 h, 4°C) and washed sequentially with low salt buffers supplemented with 0.1% NP-40 and a final wash in 1×TE buffer. The chromatin was released from the Protein A Sepharose using 1% SDS, 0.1 M NaHCO$_3$, treated with proteinase K, extracted with TE-saturated phenol/chloroform (1:1) and DNA purified using Minielute DNA columns (Qiagen, Maryland, USA). DNA (20–100 ng) was quantified using Quant-ITTM PicoGreen and validated for purity and size using a High Sensitivity DNA chip (Agilent 2100 Bioanalyser; Agilent Technologies, Santa Clara, CA, USA). DNA samples (10 ng) isolated from the immuno-purified nucleosomes and input nucleosomes were used to generate sequencing libraries for each sample using the Illumina TruSeq ChIP-Seq kit and these were sequenced using the Illumina HiSeq2000 (H3K27ac and H3K4me3) platform (Illumina, San Diego, CA, USA) and Illumina Genome Analyser II (GAII) for H3K27me3. Approximately 205 M 50 bp sequence reads per sample were produced. Reads were mapped to the unmasked ovine genome sequence (Ovis aries Oar_v3.1.74) using the NGS core tool mapping application in CLCBIO (mapping parameters: length fraction = 0.7; similarity fraction = 0.8; penalties, mismatch = 2, insertion = 3, deletion = 3). A range of 75–90% of reads was uniquely mapped and retained. Peak calling comparing the H3K4me3 or H3K27ac ChIP-Seq versus the input control was performed using MACS[75]. Only peaks found in both replicates per chromatin mark, either H3K4me3, H3K27ac or H3K27me3, were further considered. For H3K4me3 there were 16098 peaks with a FDR 0.1%, whereas for H3K27ac there were 35622 peaks with 1% FDR and for H3K27me3 31942 peaks with 5% FDR. The number of peaks is comparable to those previously reported for the same chromatin marks in cattle[76].

**Validation of the predicted sheep epigenome using ChIP-seq.** First, we calculated the overlap between experimentally observed (ChIP-Seq) sheep enhancer and promoter elements with our collection of predicted epigenome features. We created 1000 randomisations for each genomic feature using bedtools shuffle (-noOverlapping)[77]. We then calculated overlap for each randomisation against the experimentally observed (ChIP-Seq) sheep enhancer and promoter elements. Finally, we counted how many times an equal or greater overlap observed in the original features were observed in the 1000 randomisations to estimate an empirical p-value per overlap and feature.

**Epigenome feature enrichment within selective sweep regions.** The Bioconductor package Locus Overlap Analysis (LOLA) was used to assess the relationship between selection sweeps and genome regions containing functional annotation[49]. This process used (i) a 'query set' comprising each genome feature derived from four sources as detailed in Fig. 3a; (ii) a 'reference set' of 1420 genomic bins with evidence of selection (Fig. 2a) and (iii) a 'Universe Set' containing all 20 kb genome-wide bins. A Fisher's exact test with false discovery rate correction was performed to assess the significance of overlap in each pairwise comparison[49].

**Analysis of divergent allele frequencies.** Allele frequency (AF) was estimated for each SNP separately for domestic and wild sheep genomes using PLINK V1.9 (--freq −within)[71]. Next, the difference (ΔAF) was used to identify SNP with divergent AF between species. SNP was allocated into ΔAF bins and compared across functional annotation features by using bedtools intersect[77]. Log2 fold change of the observed SNP count for each genomic feature in each bin was compared against the expected SNP count (M-value; average across bins)[8,50]. Statistical significance of the deviations from the expected values was tested with a chi-squared ($\chi^2$) test.

**Data availability.** Genome sequence for each of 68 domestic sheep are available through the International Sheep Genomics Consortium (http://www.sheephapmap.org/) with Fastq files deposited in the SRA repository (https://www.ncbi.nlm.nih.gov/bioproject/160933) and SheepGenomes DB project (http://sheepgenomesdb.org/) via European Variant Archive (EVA) study PRJEB14685 (http://www.ebi.ac.uk/ena/data/view/PRJEB14685). Genome sequence for 19 mouflon (Ovis orientalis) were made available by the NEXTGEN project (http://nextgen.epfl.ch/) with Fastq files deposited in the ENA public repository as study number PRJEB3139 and secondary accession ERP001583 (https://www.ebi.ac.uk/ena/data/view/PRJEB3139). Fetal Perirenal adipose tissue H3K27ac, H3K4me3 ChIP-Seq data are accessible in the GEO series (GSE90812). Human epigenomic tracks converted to sheep coordinates are available through CSIRO Data Access Portal https://doi.org/10.4225/08/5a03a9c39a0ba.

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

## Acknowledgements

M.N.-S. and Q.N. are funded by the CSIRO Science Excellence Research Office. M.P.-E. was funded by a CSIRO McMaster Visiting Fellowship. Mouflon genome sequences were generated by the NextGen Consortium, as part of the European Union's Seventh Framework Programme (FP7/2010-2014 grant 244356). We thank the Baylor College of Medicine Human Genome Sequencing Center production teams, including S.L. Lee, S. Dugan, S. Jhangiani, D. Bandaranaike, M. Batterton, M. Bellair, C. Bess, K. Blankenburg, H. Chao, S. Denson, H. Dinh, S. Elkadiri, Q. Fu, B. Hernandez, D. Kalra, M. Javaid, J.C. Jayaseelan, S. Lee, M. Li, X. Liu, T. Matskevitch, M. Munidasa, R. Najjar, L. Nguyen, F. Ongeri, N. Osuji, L. Perales, L.-L. Pu, M. Puazo, S. Qi, J. Quiroz, R. Raj, J. Shafer, H. Shen, N. Tabassum, L.-Y. Tang, A. Taylor, G. Weissenberger, K. Wilczek-Boney, Y.-Q. Wu, Y. Xin, Y. Zhang, Y. Zhu and X. Zou.

## Author contributions

M.N.-S. and J.K. conceived the experiments, analysed the data and wrote the paper. Q.N., S.W., L.R.P.-N., A.R., M.P.-E, R.B, S.C, A.C. and H.D. performed data analysis. R.T. and T.V. contributed sheep histone modification data and analysis. K.C.W., R.A.G., D.M.M. and S.N.J. contributed sequencing of domestic sheep genomes. W.Z., S.N., H.R.R., F.P. and P.T. contributed genome sequence from Asian mouflon. J.K., N.C., K.C.W., R.B. and H.D. provided project management under the auspices of the International Sheep Genomics Consortium.
