## [Peer Review File · Nature Communications]

Reviewers' comments:

Reviewer #1 (Remarks to the Author):

This manuscript reported whole genome re-sequencing of 70 animals from 17 wild sheep and 43 domestic breeds. Based on these data, authors further examined the underlying changes during domestications. They also presented functional annotation of the sheep genome through the use of human data and predicted ovine regulatory elements based on RoadMap Epigenome and ENCODE human genomic features. Domestication of plants and animals is an interesting topic. My major concerns are listed follow:

1. This paper only presented candidate genes associated with sheep domestication processes. However, the connection of sheep domestication with human history was left out.
2. The divergence between breeds and their migration histories were not discussed.
3. I am wondering the reliability of the functional annotation and predicted ovine regulatory elements based on RoadMap Epigenome and ENCODE human genomic features. Would you please provide more bases for doing so?
4. In addition, annotation and comparison seem to be too more: (1) you can put them in the sup. note or (2) publish them in the other journal. IN the current situation, it is not coherent with the domestication line.
5. How do you assure the accuracy of inferring comparative functional annotation, the experiments verify using ChIP-Seq of adipose tissue?
6. Another concern is the sensitivity of the method to identify various type of regulatory elements.
7. Please give more details in the sup. note about the robustness of the pipeline of comparative functional annotation, the result of reciprocal liftOver program and other important temporary files. The authors also need to provide extra experimental proof.
8. It is true that many complex trait is often governed by changes in gene expression rather than nonsynonymous variations in protein sequence. In the conclusion and in the introduction, the authors are too affirmative about remodeling gene expression drove phenotypic diversification during sheep domestication without any evidence. Actually, it is not surprising that selective sweeps regions are significantly enriched for regulatory elements. The key point: which regulatory elements were located in the sweeping region? How did you determine the regulatory variants in the sweeping regions between domestic and wild sheep? Do these regulatory variants have distinct morphological correlations?
9. Can you provide more evidence for morphological change in response to the transcription variations in sheep?

Reviewer #2 (Remarks to the Author):

This study used the human functional annotation information to annotate sheep genome, followed by ChIP-Seq validation of two chromatin modifications (H3K27ac and H3K4me3) in sheep brown adipose tissue. At the same time by re-sequencing 67 sheep individuals representing 43 global breeds and 17 mouflon sheep which is the wild ancestor of domestic sheep, the authors identified selective sweeps in domestic sheep. Nicely through imposing the sweeps onto the functional annotations, the study observed that both protein coding genes and proximal regulatory elements played important roles during the evolution of modern sheep, and particularly the observation that elements immediately adjacent to genes rather than distal have played important roles in domestication and selection of sheep is a very interesting discovery. The results as well as the data of this study bear important

scientific significances not only to the sheep research community but also to the general biology. It represents the leading frontiers to understand the genetic mechanism underpinning domestication as well as the relative roles between CDS and regulatory elements in the evolution of phenotypic traits. My major and minor detailed comments are as followings.

Major comments:

1. It is a leading try to evaluate the selection pressure on regulatory elements. However, the selection sweeps showed an average size ~36.3 kb in this study, whereas the tiny proximal regulatory features were very close to their gene bodies. Therefore, it is unlikely to distinguish the enrichment of selection sweeps for regulatory features or some kind functions of gene bodies. As we know, lots of artificial selected genes are transcription factor genes, which have fruitful regulatory elements themselves. I think it is very difficult to prove the proposed enrichment of selection sweeps for proximal regulatory features. My recommendation is to tone down this finding and present the hypotheses in a shortened form as an inspiration for future research.

2. It is also somewhat conflicting between the statement "sweeps are significantly enriched for proximal regulatory elements of genes and genome features associated with active transcription" and "non-synonymous substitutions were strongly enriched for SNP in high |AF bins, while no relationship was detected for introns, UTR or intergenic regions" because intergenic regions include regulatory regions. Please try to reconcile more in the related parts to make this clearer. And I also suggest to use the similar statement in Discussion "The data revealed both protein coding genes and proximal regulatory elements played major roles during the evolution of modern sheep" in the Abstract to avoid misleading audience. Based on the data the study has obtained, it would be great if the authors could discuss the relative roles between CDS and regulatory changes in domestication process.

3. For the newly identified domestic/improved genes in sheep, such as KITLG, LCORL and FBXL3, there are no functional data in sheep backing up the story. I think some extra work about RNA-seq or allele frequency comparison across different breeds could be helpful.

4. I think the statement "gene inactivation has not played a major role during sheep domestication" is not accurate, because mutations in regulatory elements and radical non-synonymous substitutions could also inactivate the genes. And there are some bias to go through all of the nonsense mutations as using the domestic sheep gene annotation. Pseudogenes with stop codons and modifying the splicing machinery may not be annotated in Oar v3.1. I would rather say "frame-changing mutations may not play a major role during sheep domestication".

5. The ms didn't explain why chose brown adipose tissue rather than other tissues to validate annotation.

6. The key words did not reflect the key points, i.e. functional annotation and proximal regulatory elements selected by human.

7. This study acquired a total of 28.1 million SNPs across domestic and mouflon genomes. However, in "Site Frequency Analysis" section, it is not clear if the study only used "14 million sites segregating in both species" or used the full 28.1 million set to get the 9,161 SNP (0.016%) had |AF higher than 0.8 and only 1059 sites had divergence approaching the maximum (> 0.9). It should be clearly clarified, as using the whole SNP sets. And the number of SNPs with |AF higher than 0.8 or 0.9 in the detailed genome features in Fig. 4b, c, and Fig. S10 should be indicated in a Supplementary table, such as Supplementary Table 14.

8. "Assessment of protein coding gene annotations derived from OARv3.1 revealed non-synonymous substitutions were strongly enriched for SNP in high | AF bins, while no relationship was detected for introns, UTR or intergenic regions (Fig. 4b)." I am curious that why synonymous substitutions were also strongly enriched for SNP in high | AF bins, in Fig. 4b.

Minor comments:

1. In abstract, in species names the genus name should be full since they are the first places, and *O. aries* should be separated by a blank.
2. There is no page number, which brought inconvenience to reviewing.
3. When claiming "domestic sheep has lower nucleotide diversity (π) than wild sheep (domestic sheep $\pi = 0.16\%$ per nucleotide, mouflon $\pi = 0.20\%$)", a statistic test should be applied and a p value should be given.
4. (LOLA62 should be (LOLA62).
5. This revealed selection sweeps \diamond These revealed selection sweeps.
6. The version of BWA, GATK, bedtools and VCFtools should be indicated.
7. "Striking enrichment was detected for chromatin states associated with both active and developmentally poised promoters (TxFlnk, TssBiv) as well as proximal enhancers (EnhG, Fig. 4b)." should be "Fig. 4c".
8. Fig. 5 a, the FBXL3 gene should also be represented in sheep coordinate.

Reviewer #3 (Remarks to the Author):

General

Naval-Sanchez et al. conduct a genome wide scan of F_{ST} in 20kb windows between domestic sheep and their wild ancestor (Mouflon), to identify loci associated with domestication. They then conduct an analysis to determine if these selective sweeps are enriched for regulatory elements, from both leftover of human ENCODE and Roadmap features and H3K27ac and H3K4me3 derived ChIP-Seq peaks from sheep brown adipose tissue.

A strong point of the paper is using two geographically isolated populations of Mouflon, to represent the ancestral sheep population, such that the regions with large F_{ST} between the wild ancestor and domestic sheep are not just the result of drift within one population of Mouflon alone.

A major concern is the precision with which the selective sweeps are mapped, and the implications for the enrichment analysis. The size of window step was 10kb – surely this includes both gene and regulatory regions (eg promoter, some enhancers) in many cases. So how can a conclusion be reached about which class of gene annotation is more important? One way of testing this would be to randomly pick a base pair within each of 430 genes, change the allele frequency in the sequenced animals so that it is substantially different between wild and domestic sheep, then rerunning the F_{ST} and enrichment analysis. If the regulatory regions come up as enriched in this analysis, it demonstrates the result is spurious. Likewise, a base pair could be randomly chosen within 430 regulatory elements across the genome, allele frequencies changed so that there is a large difference between domestic and wild sheep, and the analysis rerun. This would assess the power of the enrichment analysis.

Specific comments

Abstract (and elsewhere)

Change farm-yard to farmed

Results

Page 9 Line 186. It should be pointed out that the number of regulatory elements predicted from ENCODE and Roadmap is much larger than validated with the Histone modifications, and a reason given. One possibility is the leftover gives a lot of false positive results.

Discussion

Need to mention that precision of mapping the selective sweeps is a limitation.

Page 15 Line 348. Assume you mean domestication here rather than evolution, given the focus of the paper and the results?

Dear Editor,

Please find below our responses to the review process. The comments of each Reviewer were valuable, and assisted us craft an interpretation that better reflects the strengths and limitations of the dataset. We have performed additional data collection and analysis, largely in response to the comments of Reviewer#1, and implemented a number of other changes throughout. These are described in detailed below.

On behalf of the authors I look forward to learning if the manuscript will move forwards towards publication.

Best Regards, James

Reviewers' comments:

Reviewer #1 (Remarks to the Author):

This manuscript reported whole genome re-sequencing of 70 animals from 17 wild sheep and 43 domestic breeds. Based on these data, authors further examined the underlying changes during domestication. They also presented functional annotation of the sheep genome through the use of human data and predicted ovine regulatory elements based on RoadMap Epigenome and ENCODE human genomic features. Domestication of plants and animals is an interesting topic. My major concerns are listed follow:

1. This paper only presented candidate genes associated with sheep domestication processes. However, the connection of sheep domestication with human history was left out.

Authors Response:

The authors agree that the connection between human historical migration and the spread and evolution of their domestic animals is of interest. We briefly mention in the introduction that *“The domestication of plants and animals commenced around ten thousand years ago and precipitated enormous societal change by transitioning human king from hunter-gatherers to agricultural settles^{1,2}”*

Regrettably it was beyond the scope of the current study to explore this topic in detail.

2. The divergence between breeds and their migration histories were not discussed.

Authors Response:

Our experimental design captured a broad selection of breeds, but only 1 or 2 animals per breed. As a result, our view is that divergence between breeds can't be accurately assessed when so few individuals are being used to represent the diversity present within a given breed. This is especially true for sheep which exhibit large within breed diversity. These types of studies are best approached using SNP arrays deployed across 50 – 100 animals per population. It was therefore not addressed in this study which had different objectives.

3. I am wondering the reliability of the functional annotation and predicted ovine regulatory

elements based on RoadMap Epigenome and ENCODE human genomic features. Would you please **provide more bases for** doing so?

Authors Response:

The basis for exploiting available human data is strong, with the rationale consists of the following points:

i) Sequence conservation of noncoding elements between diverse species has been successfully used to identify functional regulatory sequences in multiple studies³⁻⁷. This serves as an important precedent to support the experimental approach described in the manuscript. For example, independent functional annotation of human and mouse via the ENCODE and Mouse ENCODE consortia facilitated comparative analyses of regulatory regions⁸⁻¹¹. Using reciprocal blastz to compare the functional evolution of mouse-human homologous regions found 44% of promoters and 40% of enhancers in mice that mapped to human retained the same cell/tissue functional activity. Higher rates were observed when compared to datasets derived using multiple tissues. We have employed the same reciprocal blast methodology that relies on the conservation of regulatory elements at the DNA sequence and genome organizational levels (i.e. location in the genome).

ii) The human ENCODE, FANTOM, ROADMAP and related projects have generated large volumes of data relevant to the identification of promoters, enhancers and other regulatory DNA elements. The information generated in the foreseeable future for sheep is likely to remain far less comprehensive for the number of tissues, sampling conditions and breadth of annotation of regulatory elements compared to that identified for human and mouse.

iii) We evaluated the reliability of the approach using multiple methods. The detail of these validation steps are addressed below in response to other points.

In order to more clearly articulate for the reader the basis for proceeding to build and utilise a comparative annotation, we have added the following sentences into the introduction:

“Encouragingly, sequence conservation of noncoding elements has been successfully used to identify functional regulatory sequences between diverse species²⁰⁻²⁴. Further, recent comparative genomic studies in mammals have shown that despite a significant divergence of Transcription Factor Binding Sites (TFBSs)^{25,26} and enhancers^{27,28}, there is a substantial core of regulatory elements which can be characterized based on sequence conservation²⁹⁻³¹. For example, Mouse ENCODE comparative analyses concluded that 44% of promoters and 40% of enhancers in mice that mapped to human retained functional and tissue specific activity³⁰.”

4. In addition, annotation and comparison seem to be too more: (1) you can put them in the sup. note or (2) publish them in the other journal. IN the current situation, it is not coherent with the domestication line.

Authors Response:

Thank you to the reviewer for his/her comment, however we believe that the annotation of regulatory features is elemental to the manuscript, and should therefore be retained. To focus on their application to progress our understanding of selection and domestication, the bulk of the results relating to the annotation and it's assessment are within the Supplementary Material.

5. How do you assure the accuracy of inferring comparative functional annotation, the experiments verify using ChIP-Seq of adipose tissue?

Authors Response:

We applied a variety of approaches to assess the accuracy with which the methodology predicted functional elements. The reviewer is correct that one of the major approaches focussed on the experimental collection of ChIP-Seq data from sheep tissue to facilitate comparison with feature prediction. We recovered 85% (promoters) and 71% (enhancers) of the experimentally identified features, leading us to conclude the predictive approach returned a level of accuracy suitable to support the subsequent analysis presented in the manuscript. The approaches used were:

i) Dedicated collection of histone modification data from sheep tissue and overlap analysis against our predicted promoter and enhancer collections (Figure 3b, 3c). We performed overlap analysis against other feature classes and compared the result to features randomly selected in the sheep genome (Supplementary Figures 4-6).

ii) Physical proximity testing of genome features with TSSs (Supp. Figure 8). This sought to validate the expected colocation between predicted promoter elements and TSSs.

iii) Nucleotide diversity estimates compared between genome features (Supp. Figure 9). This sought to inspect expected differences in evolutionary constraint between predicted feature classes.

iv) Nucleotide diversity estimates in elements with different expression ranges (Supp. Figure 10). This sought to evaluate expectations about constraint as a function of tissue expression ubiquity.

v) Recovery percentages across different feature types were collated (Supp. Figure 5).

We recognise that reliance on a comparative methodology is likely to introduce bias. To ensure this is made clear for the reader, we have included the following cautionary text into the discussion:

“The reciprocal BLAST methodology may have introduced bias, resulting in an over-representation of evolutionarily conserved promoter elements compared with enhancers. Indeed the recovery of enhancers within sheep ChIP-Seq data was lower than for promoters, however the difference was not large and enhancers have a higher evolutionary turnover compared to proximal regulatory elements or protein-coding sequences. Further, it is possible that species specific and tissue specific regulatory elements are likely to be under represented as a consequence of the methodology used.”

6. Another concern is **the sensitivity** of the method to identify **various types of regulatory elements**.

Authors Response:

It is possible the bias referred to above manifests through sensitivity differences to detect different types of regulatory elements. We shared the concern raised by the reviewer, which prompted the following approaches:

i) when designing the ChIP-Seq data collection from sheep, we selected for two histone modification marks that identify separate types of regulatory elements (promoters and enhancers). This facilitated analysis to assess the success of feature prediction separately for the different feature types. The result is presented in Figure 3c, and demonstrated an acceptable level of sensitivity to discriminate between promoters and enhancers.

ii) a key piece of evidence supporting our approach was derived from the extent to which our predicted elements were validated by direct experimental observation using ChIP-Seq. To evaluate the strength of any overlap, we estimated the expected overlap by chance. This is recorded in Supplementary Table 10 for each annotation feature, and assists the reader calibrate the strength of overlap observed and the differences between them.

iii) to more fully address the question of sensitivity we collected a third histone modification mark from sheep tissue and incorporated the results into the revised manuscript. We selected H3K27me3 as it identifies Polycomb repression, a very different type of regulatory element to either promoters or enhancers. Pleasingly, overlap analysis and comparison to random expectation revealed strong enrichment in our predicted feature set with states that included Repressed Polycomb (ReprPC), Bivalent/Poised TSS (TssBiv) and Flanking bivalent TSS/Enh (BivFlnk). This is captured in a revised version of Supplementary Table 10, and confirms that our approach had sensitivity to discriminate repressive regulatory features. The new H3K27me3 ChIP-seq data sheep data has been uploaded as GEO series (GSE90812).

The following text has been added to manuscript and the methods updated:

“Next, we performed H3K27me3 ChIP-seq in brown adipose tissue sheep, a mark related to Polycomb repression. This yielded 31942 elements in sheep. Comparison against the set of predicted regulatory features revealed strong enrichment only for repressive features including repressed Polycomb and bivalent / poised TSSs (Supplementary Table 10). This provides confidence our comparative approach successfully recovered biologically meaningful genome feature annotations with the sensitivity to discriminate promoters, enhancers and repressed regulatory elements.”

7. Please give more details in the **sup. note** about the **robustness of the pipeline** of comparative functional annotation, the result of reciprocal liftOver program and other important temporary files. The authors also need to **provide extra experimental proof**.

Authors Response:

The liftOver pipeline is one of the key methodologies in the manuscript, so we agree with the reviewer that additional analysis, a more detailed explanation and the availability of temporary files is important. In response we have made the following changes:

i) To address the question of robustness, we performed additional analysis and provide new results that test if the liftOver pipeline is highly sensitive to the particular combination of species in which it is applied (ie human and sheep). We applied our liftOver pipeline to convert human Roadmap Epigenomics chromHMM TssA and Enh data into mouse (mm10), pig (susScr3) and cow (bosTau6). The predicted functional elements in each genome were then assessed for enrichment within species-

specific experimentally derived H3K4me3 regions (promoters) and enhancers (regions with H3K27ac signal not overlapping H3K4me3 regions)¹⁷. This revealed that, as in sheep, liftOver of human promoters and enhancers from 127 epigenomes successfully recovered high proportions of mouse, pig and cow functional elements. The result of this new analysis is described in Supplementary Table 11 and a new Supplementary Figure (11). A short description has been added into the results section of the main text.

Supplementary Figure 11.

ii) More detail was provided in the Supplementary information, as part of the expanded description of Supplementary Figure 11. It reads:

“The comparative approach used in this study employs liftOver genome coordinates conversion¹. This is based on chain and net files between genome assemblies. Chains refer to ordered sequence of pairwise nucleotide alignments by BLASTZ 2 separated by large gaps. Nets are a higher-order alignments, which track which bases are covered by more than one chain as well as annotating inversions and or other displacements between genome assemblies. Nets are not symmetric, meaning they show the best alignment of one species versus the other with the consequence they can differ in the reciprocal net. To optimise for the best alignment from both species we prepare a “reciprocal best” alignment. It is noteworthy that this method does not allow us to identify lineage-specific duplications.”

(iii) The reviewer requests the results of reciprocal liftOver program and temporary files. The following files have been added to the Supplementary information:

a) Sheep epigenome predicted features.tar.gz: contains the final reciprocal best alignment from ENCODE proximal as well as chromHMM ROADMAP features. The result of reciprocal liftOver.

b) liftOver sheep temporary files.tar.gz: We have added a new tar file with liftOver temporary files containing i) LiftOver temporary files mapping human to sheep, ii) LiftOver temporary files mapping sheep back to human and iii) Dictionary files containing the link between human to sheep coordinates for exact best-reciprocal files.

8. It is true that **many complex trait** is often governed **by changes in gene expression** rather than nonsynonymous variations in protein sequence. In the conclusion and in the introduction, the authors are **too affirmative about** remodeling gene expression drove phenotypic diversification during sheep domestication without any evidence. Actually, it is not surprising that selective sweeps regions are significantly enriched for regulatory elements. The key point: **which regulatory elements were located in the sweeping region? How did you determine the regulatory variants in the sweeping regions between domestic and wild sheep? Do these regulatory variants have distinct morphological correlations?**

Authors Response:

The reviewer has posed a series of questions in comment 8 that we address in point form below:

‘the authors are **too affirmative** about remodeling gene expression drove phenotypic diversification during sheep domestication without any evidence.’

Authors Response:

This was raised by Reviewer#2 in light of the potentially confounding effect of linkage disequilibrium. We have modified the associated statements to be less affirmative. Details are provided in response to Reviewer #2 comment 1.

‘The key point: **which regulatory elements** were located in the sweeping region?’

Authors Response:

We evaluated 29 genomic features for their enrichment in selection sweeps, as presented in Figure 4a. One selective sweep may overlap with distinct regulatory elements. However, we found proximal promoters, transcription at the ends of genes (TxFlnc) and actively transcribed states (Tx) were statistically enriched for the set of selective sweeps here identified. This is described in Figure 4.

‘How did you determine the regulatory variants in the sweeping regions between domestic and wild sheep?’

Authors Response:

The methods used to identify sweep regions is detailed in the methods paragraph “*Genome Scanning for selection sweeps*”, while the approach implemented to determine the extent of overlap with regulatory regions is given in the section “*Co-location analysis of selective sweep regions and epigenome features*”. Both methods sections provide a detailed description and need not be replicated here.

‘Do these **regulatory variants** have **distinct morphological** correlations?’

Authors Response:

The authors agree that it would be beneficial to establish links between the variants and their morphological consequence. This was also raised by Reviewer #2 as comment 3, and our response is

given against this more detailed point.

9. Can you provide **more evidence for morphological change** in response **to the transcription variations** in sheep?

Authors Response:

The manuscript does not contain transcriptional data. As a result, we have elected not to provide additional commentary.

Reviewer #2 (Remarks to the Author):

This study used the human functional annotation information to annotate sheep genome, followed by ChIP-Seq validation of two chromatin modifications (H3K27ac and H3K4me3) in sheep brown adipose tissue. At the same time by re-sequencing 67 sheep individuals representing 43 global breeds and 17 mouflon sheep which is the wild ancestor of domestic sheep, the authors identified selective sweeps in domestic sheep. Nicely through imposing the sweeps onto the functional annotations, the study observed that both protein coding genes and proximal regulatory elements played important roles during the evolution of modern sheep, and particularly the observation that elements immediately adjacent to genes rather than distal have played important roles in domestication and selection of sheep is a very interesting discovery. The results as well as the data of this study bear important scientific significances not only to the sheep research community but also to the general biology. It represents

the leading frontiers to understand the genetic mechanism underpinning domestication as well as the relative roles between CDS and regulatory elements in the evolution of phenotypic traits. My major and minor detailed comments are as followings.

Major comments:

1. It is a leading try to evaluate the selection pressure on regulatory elements. However, the selection sweeps showed an average size ~36.3 kb in this study, whereas the tiny proximal regulatory features were very close to their gene bodies. Therefore, it is unlikely to distinguish the enrichment of selection sweeps for regulatory features or some kind functions of gene bodies. As we know, lots of artificial selected genes are transcription factor genes, which have fruitful regulatory elements themselves. I think it is very difficult to prove the proposed enrichment of selection sweeps for proximal regulatory features. My recommendation is to tone down this finding and present the hypotheses in a shortened form as an inspiration for future research.

Authors Response:

The authors agree that selection sweep methodology alone makes it difficult to tease apart the contributions of multiple co-located genomic features such as proximal regulatory elements and coding exons. It was precisely this limitation that prompted inclusion of the site frequency analysis, which is independent of the definition of sweep regions and evaluates every nucleotide separately. None the less, the authors agree that 'toning down' the emphasis is prudent. In response we have

implemented the following revisions:

i) We have modified the abstract to be less assertive as follows:

“Our data demonstrates that remodelling of gene expression was an important component of the evolutionary forces”

Has been modified to read:

“Our data demonstrates that remodelling of gene expression is likely to have been one of the evolutionary forces”

ii) We have added sentences into the discussion that serve as a direct cautionary note to readers concerning the impact of LD and the associated interpretation of the data as follows:

“A final challenge arises from linkage disequilibrium, blurring the relative contribution of physically co-located genomic features. A case in point is the observation that synonymous sites displayed enrichment in our site frequency analysis. It is not clear if this reflects a functional role for many synonymous positions via cryptic effects on transcription or mRNA transport as previously suggested^{126,127} or if their proximity to missense mutations has generated the enrichment via linkage disequilibrium.”

2. It is also somewhat conflicting between the statement “sweeps are significantly enriched for proximal regulatory elements of genes and genome features associated with active transcription” and “non-synonymous substitutions were strongly enriched for SNP in high bins, while no relationship was detected for introns, UTR or intergenic regions” because intergenic regions include regulatory regions. Please try to reconcile more in the related parts to make this clearer. And I also suggest to use the similar statement in Discussion “The data revealed both protein coding genes and proximal regulatory elements played major roles during the evolution of modern sheep” in the Abstract to avoid misleading audience. Based on the data the study has obtained, it would be great if the authors could discuss the relative roles between CDS and regulatory changes in domestication process.

Authors Response: The reviewer makes a good point. Our interpretation is that regulatory regions represent only a fraction of the intergenic genomic feature, potentially explaining the apparent contradiction. To address we have added the following sentence:

“Regulatory elements are embedded within the intergenic genome feature, however they likely represent an insufficiently large proportion to cause intergenic regions to display enrichment.”

To address the bottom component of the reviewer’s comments, we have modified the abstract as requested to reflect the text in the discussion. In our view, this removes an impression that coding regions were not identified as important. The abstract now reads:

“we evaluate the impact of selection and domestication on regulatory sequences and find that sweeps are significantly enriched for protein coding genes, proximal regulatory elements of genes and genome features associated with active transcription”

3. For the newly identified domestic/improved genes in sheep, such as KITLG, LCORL and FBXL3,

there are no functional data in sheep backing up the story. **I think some extra work about RNA-seq or allele frequency comparison across different breeds could be helpful.**

Authors Response:

The reviewer raises a good point, and we agree that functional data would be beneficial. Our view is that dedicated GWAS using animal populations displaying variation in the phenotypic traits controlled by these genes is the best way to understand their functional consequence. This was beyond the scope of the current experiment. Allele frequency data across large numbers of animals was not available as the variants are not present on SNP arrays, and RNA-seq alone is likely to be of restricted value unless until additional information is available describing the tissues and time points relevant to sample.

To assist in making this clear for the reader, we have added a sentence into the results to more clearly define the boundary of the current work:

“Each of these candidate genes is likely to contribute to phenotypic variation, however in the absence of individually recorded trait data analysed using an approach such as GWAS, establishing the direct link between these genes and their functional consequence is difficult using selection sweep methodology.”

4. I think the statement “gene inactivation has not played a major role during sheep domestication” is not accurate, because mutations in regulatory elements and radical non-synonymous substitutions could also inactivate the genes. And there are some bias to go through all of the nonsense mutations as using the domestic sheep gene annotation. Pseudogenes with stop codons and modifying the splicing machinery may not be annotated in Oar v3.1. I would rather say “frame-changing mutations may not play a major role during sheep domestication”.

Authors Response:

The authors agree the statement is not accurate. We have modified in accordance with the reviewer’s suggestion and it now reads:

“our conclusion is that gene inactivation via nonsense (frame-changing) and splice site mutation has not played a major role during sheep domestication.”

5. The ms didn’t explain why chose brown adipose tissue rather than other tissues to validate annotation.

Authors Response:

Brown adipose tissue is important in a production setting for the sheep industry, as it influences lamb survival under some circumstances and may be predictive of other performance traits. We are not aware of any reason why it would be either more or less relevant for validating our annotation results. In response to the reviewer’s comments we have:

- added text into the Methods section supporting why adipose was selected
- added a note of caution in the discussion section that reads:

“Further, it is possible that species specific and tissue specific regulatory elements are likely to be under represented as a consequence of both our methodology and the application of ChIP-Seq using only a single sheep tissue.”

6. The key words did not reflect the key points, i.e. functional annotation and proximal regulatory elements selected by human.

Authors Response:

Implemented. The key words suggested have been added.

7. This study acquired a total of 28.1 million SNPs across domestic and mouflon genomes. However, in “Site Frequency Analysis” section, it is not clear if the study only used “14 million sites segregating in both species” or used the full 28.1 million set to get the 9,161 SNP (0.016%) had $\hat{A}F$ higher than 0.8 and only 1059 sites had divergence approaching the maximum (> 0.9). It should be clearly clarified, as using the whole SNP sets. And the number of SNPs with $\hat{A}F$ higher than 0.8 or 0.9 in the detailed genome features in Fig. 4b, c, and Fig. S10 should be indicated in a Supplementary table, such as Supplementary Table 14.

Authors Response:

This wasn't clear in the text. The proportions provided were derived from 14 M SNP that segregate in both species, and not from the larger collection of 28.1 M found in either species. The text has been edited to clarify this for the reader as follows:

“We first estimated allele frequencies within domestic and wild sheep using 14 M sites found to be segregating in both species.”

“.. with 8.9 M sites (63% of 14 M, Supplementary Table 13) having allele frequency difference (delta AF) less than 0.2.”

In order to perform the site frequency analysis we required SNPs with highly reliable allele frequency estimates as in ¹⁹. Therefore, in order to avoid artefacts we performed the site frequency analysis in SNPs with a MAF >0.05 disregarding rare alleles which are prone to underlie sequencing artefacts.

8. “Assessment of protein coding gene annotations derived from OARv3.1 revealed non-synonymous substitutions were strongly enriched for SNP in high $\hat{A}F$ bins, while no relationship was detected for introns, UTR or intergenic regions (Fig. 4b).” I am curious that why synonymous substitutions were also strongly enriched for SNP in high $\hat{A}F$ bins, in Fig. 4b.

Authors Response: We are curious too. There are two possibilities. First, that synonymous sites are under higher evolutionary pressure than generally assumed and have been enriched in high ΔAF bins as a result. There is precedence for this suggestion, whereby synonymous sites have a proven role on function via effects on transcription and/or mRNA transport. We have included reference to this as detailed below in the modified text. Secondly, it may be due to the influence of linkage disequilibrium as described elsewhere in our responses. Unfortunately we are not, at this time, able to discriminate between the two possibilities. To address this point of interest, the following text has been inserted into the discussion section:

“A case in point is the observation that synonymous sites displayed enrichment in our site frequency analysis. It is not clear if this reflects a functional role for many synonymous positions via cryptic effects on transcription or mRNA transport as previously suggested ^{126,127} or if their proximity to missense mutations has generated the enrichment via linkage disequilibrium”.

Minor comments:

1. In abstract, in species names the genus name should be full since they are the first places, and *O. aries* should be separated by a blank.

Authors Response: Corrected.

2. There is no page number, which brought inconvenience to reviewing.

Authors Response: Noted. The journal's guide to submission doesn't request they be included.

3. When claiming "domestic sheep has lower nucleotide diversity (π) than wild sheep (domestic sheep $\pi = 0.16\%$ per nucleotide, mouflon $\pi = 0.20\%$ ", a statistic test should be applied and a p value should be given.

Authors Response: Implemented. We applied the Wilcoxon rank sum test suitable for comparing two matched sample distributions. Using the pi value for all windows, the test returned a significant result, confirming the claim made. The associated p-value has been added into the text.

4. (LOLA62 should be LOLA62).

Authors Response: Corrected.

5. This revealed selection sweeps à These revealed selection sweeps. **Authors Response: Edited to now read: "This approach revealed selection sweeps..."**

6. The version of BWA, GATK, bedtools and VCFtools should be indicated.

Authors Response: Implemented. The versions for each package have been inserted.

7. "Striking enrichment was detected for chromatin states associated with both active and developmentally poised promoters (TxFlnk, TssBiv) as well as proximal enhancers (EnhG, Fig. 4b)." should be "Fig. 4c".

Authors Response: Thanks for picking this up. Corrected.

8. Fig. 5 a, the FBXL3 gene should also be represented in sheep coordinate.

Authors Response: Implemented. The sheep genome coordinates have been added to the Figure.

Reviewer #3 (Remarks to the Author):

General

Naval-Sanchez et al. conduct a genome wide scan of FST in 20kb windows between domestic sheep and their wild ancestor (Mouflon), to identify loci associated with domestication. They then conduct an analysis to determine if these selective sweeps are enriched for regulatory elements, from both liftover of human ENCODE and Roadmap features and H3K27ac and H3K4me3 derived CHIP-Seq peaks from sheep brown adipose tissue.

A strong point of the paper is using two geographically isolated populations of Mouflon, to represent the ancestral sheep population, such that the regions with large FST between the wild ancestor and domestic sheep are not just the result of drift within one population of Mouflon alone.

A major concern **is the precision** with which the selective sweeps are mapped, and the implications for the enrichment analysis. The size of window step was 10kb – surely this includes both gene and regulatory regions (eg promoter, some enhancers) in many cases. **So**

how can a conclusion be reached about which class of gene annotation is more important?

One way of testing this would be to randomly pick a base pair within each of 430 genes, change the allele frequency in the sequenced animals so that it is substantially different between wild and domestic sheep, then rerunning the FST and enrichment analysis. If the regulatory regions come up as enriched in this analysis, it demonstrates the result is spurious. Likewise, a base pair could be randomly chosen within 430 regulatory elements across the genome, allele frequencies changed so that there is a large difference between domestic and wild sheep, and the analysis rerun. This would assess the power of the enrichment analysis.

Authors Response: The point raised, concerning the ability to discriminate between co-localised genomic elements using sweep methodology, is the same as raised by Reviewer #2 in point 1. It has been addressed above, with our response being that this concern prompted the inclusion of the site by site analysis. This is independent of linkage disequilibrium, which is often the constraint restricting increased precision in selection sweep methodologies. However, to address which class of gene annotation is important, we felt it would enhance the results if we more clearly indicated the outcome if random genomic bins were used (as opposed to bins identified by sweep methodologies). In response we:

i) defined 1420 20KB regions with 10KB overlap at random before assessing their genomic feature enrichment using our pipeline. There is no a significant enrichment for any feature.

ii) we include these results as Supplementary Table 12.

iii) we updated Figure 4a to include the threshold expected at random, as horizontal red bars for each of the genome features classes evaluated. This allows the reader to visually assess the enrichment (or otherwise) specific to each feature type.

Specific comments

Abstract (and elsewhere)

Change farm-yard to farmed

Authors Response: The term 'farm-yard' animals draws a distinction between livestock (eg cattle, chickens, pigs and sheep) and other species farmed for agricultural production (eg salmon, shrimp, etc). We have therefore changed "farm-yard" to "livestock" throughout.

Results

Page 9 Line 186. It should be pointed out that the number of regulatory elements predicted from ENCODE and Roadmap is much larger than validated with the Histone modifications, and a reason given. One possibility is the liftover gives a lot of false positive results.

Authors Response: Implemented. We acknowledge the point and provide an interpretation in the text inserted at page 9, line 186 as follows:

"A large number of predicted promoters were not present, possibly reflecting out CHiP-Seq data originates from only one sheep tissue and timepoint."

Discussion

Need to mention that precision of mapping the selective sweeps is a limitation.

Authors Response: Implemented. The limitations of selection sweep methodology have been made clear in the following sentences:

"establishing the direct link between these genes and their functional consequence is difficult using selection sweep methodology alone."

"A final challenge arising from linkage disequilibrium, blurring the relative contribution of of physically co-located genomic features. A case in point is the observation that synonymous sites displayed enrichment in our site frequency analysis. It is not clear if this reflects a functional role for many synonymous positions via cryptic effects on transcription or mRNA transport as previously suggested^{126,127} or if their proximity to missense mutations has generated the enrichment via linkage disequilibrium".

Page 15 Line 348. Assume you mean domestication here rather than evolution, given the focus of the paper and the results?

Authors Response: The term 'evolution' was carefully chosen as it refers to both the process of domestication and the subsequent process of artificial selection. As a result, we have elected not to substitution out the term 'evolution'.

Bibliography

1. Diamond, J. Evolution, consequences and future of plant and animal domestication. *Nature* **418**, 700–707 (2002).
2. Larson, G. *et al.* Current perspectives and the future of domestication studies. *Proc. Natl. Acad. Sci.* **111**, 6139–6146 (2014).
3. Hardison, R. Conserved noncoding sequences are reliable guides to regulatory elements. *Trends Genet.* **16**, 369–372 (2000).
4. Bejerano, G. *et al.* Ultraconserved Elements in the Human Genome. *Science* **304**, 1321–1325 (2004).
5. Siepel, A. *et al.* Evolutionarily conserved elements in vertebrate, insect, worm, and yeast genomes. *Genome Res.* **15**, 1034 (2005).
6. Woolfe, A. *et al.* Highly Conserved Non-Coding Sequences Are Associated with Vertebrate Development. *PLoS Biol.* **3**, (2005).
7. Pennacchio, L. A. *et al.* In vivo enhancer analysis of human conserved non-coding sequences. *Nature* **444**, 499–502 (2006).
8. Yue, F. *et al.* A Comparative Encyclopedia of DNA Elements in the Mouse Genome. *Nature* **515**, 355 (2014).
9. Vierstra, J. *et al.* Mouse regulatory DNA landscapes reveal global principles of cis-regulatory evolution. *Science* **346**, 1007–1012 (2014).
10. Cheng, Y. *et al.* Principles of regulatory information conservation between mouse and human. *Nature* **515**, 371–375 (2014).

11. Pervouchine, D. D. *et al.* Enhanced transcriptome maps from multiple mouse tissues reveal evolutionary constraint in gene expression. *Nat. Commun.* **6**, 5903 (2015).
12. ENCODE Project Consortium. An integrated encyclopedia of DNA elements in the human genome. *Nature* **489**, 57–74 (2012).
13. Roadmap Epigenomics Consortium *et al.* Integrative analysis of 111 reference human epigenomes. *Nature* **518**, 317–330 (2015).
14. Ernst, J. & Kellis, M. ChromHMM: automating chromatin-state discovery and characterization. *Nat. Methods* **9**, 215–216 (2012).
15. Hinrichs, A. S. *et al.* The UCSC Genome Browser Database: update 2006. *Nucleic Acids Res.* **34**, D590 (2006).
16. Schwartz, S. *et al.* Human–Mouse Alignments with BLASTZ. *Genome Res.* **13**, 103–107 (2003).
17. Villar, D., Flicek, P. & Odom, D. T. Evolution of transcription factor binding in metazoans — mechanisms and functional implications. *Nat. Rev. Genet.* **15**, 221–233 (2014).
18. García-Gómez, E. *et al.* Using regulatory and epistatic networks to extend the findings of a genome scan: identifying the gene drivers of pigmentation in merino sheep. *PloS One* **6**, e21158 (2011).
19. Carneiro, M. *et al.* Rabbit genome analysis reveals a polygenic basis for phenotypic change during domestication. *Science* **345**, 1074–1079 (2014).
20. Sheffield, N. C. & Bock, C. LOLA: enrichment analysis for genomic region sets and regulatory elements in R and Bioconductor. *Bioinformatics* **32**, 587 (2016).

Reviewers' comments:

Reviewer #1 (Remarks to the Author):

I have no further comments.

Reviewer #2 (Remarks to the Author):

The revision has addressed most of my concerns.

Reviewer #3 (Remarks to the Author):

General

The authors have responded to many of the previous comments satisfactorily. However the permutation test they have conducted in an attempt to demonstrate that they are able to distinguish between genome features which are usually in very close proximity seems inappropriate. If I understand correctly, they have chosen 1420 20kb bins at random across the genome for their permutation testing, and just tested for enrichment in those regions. This does not answer the question is their method able to distinguish between mutations in regulatory regions versus exonic regions. The correct permutation test would choose a site within the randomly selected regions, artificially change allele frequencies between wild and domestic sheep (for SNP within exons for example), and then confirm that only the exons came up as enriched. Or if sites between promoters and enhancers were altered to have extreme frequencies, no annotation should come up as enriched.

The site by site analysis is not independent of linkage disequilibrium, sites showing high divergence in allele frequencies may just be neutral sites in high linkage disequilibrium with a true causative mutation. This is the basis of "hitchhiking". Sampling error (particularly with the number of animals used here) can cause the neutral site to show more apparent divergence than the true site, unfortunately.

Specific comments.

Page 5 line 85. Identified not identifies

Page 5 line 88. Collected not collect

Page 6 Line 110. Replace and with however

Page 8. Line 167. "Two prominent examples include dorsal/ventral patterning and regulation of lipid metabolism (Table 2, Supplementary Table 7). Reduction in body size was a quick and early response to domestication across species, likely followed by human mediated regulation of muscularity and fatness as sheep were actively managed as a food source". It is not obvious how dorsal/ventral patterning is related to the sentence that follows it, please make this clear for the reader.

Page 9. Line 199. "A large number of predicted promoters were not present, possibly reflecting our ChIP-Seq data originates from only one sheep tissue and time point". Add "or alternatively, a large number of false positive hits".

Page 12. Line 274. "Only a very small fraction of loci exhibited strongly divergent allele frequency expected to be characteristic of sites underlying domestication and/or selection". A very small fraction of loci would be expected by chance to have highly divergent allele frequencies, as a result of drift alone. The authors could simulate the divergence of their populations, (using Fregene for example), and investigate if the proportion of loci showing strongly divergent allele frequency is greater than expected by chance (drift) alone.

Dear Editor,

Please find attached our second revision. It was pleasing that our earlier revision completely satisfied Reviewers #1 and #2, and we have carefully considered the remaining query from Reviewer #3. We have performed additional analyses as requested, and believe these assist us in making a balanced interpretation of our results. Our responses are highlighted in blue below.

On behalf of the authors, I look forwards to learning if the manuscript is finally acceptable,

Best Regards,

James

Reviewers' comments:

Reviewer #1 (Remarks to the Author):

I have no further comments.

Reviewer #2 (Remarks to the Author):

The revision has addressed most of my concerns.

Reviewer #3 (Remarks to the Author):

General

The authors have responded too many of the previous comments satisfactorily. However the permutation test they have conducted in an attempt to demonstrate that they are able to distinguish between genome features which are usually in very close proximity seems inappropriate. If I understand correctly, they have chosen 1420 20kb bins at random across the genome for their permutation testing, and just tested for enrichment in those regions. This does not answer the question is **their method able to distinguish between mutations in regulatory regions versus exonic regions**. The correct permutation test would choose a site within the randomly selected regions, artificially change allele frequencies between wild and domestic sheep (for SNP within exons for example), and then confirm that only the exons came up as enriched. Or if sites between promoters and enhancers were altered to have extreme frequencies, no annotation should come up as enriched.

Authors' Response:

The reviewer's concern is best captured by the sentence (in bold) that queries if the approach is "**able to distinguish between mutations in regulatory regions versus exonic regions**". To answer this, the reviewer requests an analysis which artificially alters allele frequencies in exons, before assessing the impact on multiple genome features (exons, regulatory components, etc). We agree this is a potentially informative approach, and performed the analysis as described below.

We artificially altered the allele frequency of exonic SNPs in wild and domestic sheep, such that they have extreme allele frequency differences ($\Delta AF = 0.95$ which is in the highest bin). We performed this using randomly selected regions (1420 20 Kb bins), before estimating enrichment for exons as well as a range of regulatory features. The resulting M-values reflect the strength of enrichment, for each genome feature, beyond expectation.

Allele frequency differences (Δ AF) between wild and domestic sheep following artificial manipulation of exons. To evaluate correlated effects between annotation features, exonic SNP were artificially altered to take extreme values (Δ AF = 0.95, which is in the highest bin) in a random set of 1420 20 Kb bins. The M-value reflects the strength of enrichment beyond expectation (refer to methods) and the black line gives the total number of SNPs in each Δ AF bin. Protein coding gene annotations are plotted along with 12 chromatin state annotations from the Roadmap dataset.

The result confirmed the CDS feature (exons) had extreme enrichment (M-value = 7.3), far in excess of any other feature. The 'Gene' feature type that contains both exons and introns had a much reduced M-value (1.67), confirming the method is able to distinguish between exons and their physically co-located introns. The analysis identified mild enrichment for a number of regulatory feature types including TxFlnk (transcription at gene 5' and 3'), Tx (strong transcription) and EnhG (genic enhancers). Each of these feature types physically overlap exons (TxFlnk, Tx) or are immediately adjacent (EnhG), which likely explains the mild increase in M-value. We therefore acknowledge that combinations of annotated genome feature types are not independent, but rather overlap in a complex matrix that needs to be better illuminated for the reader.

We have therefore made the following modifications to the manuscript which we feel assists with data interpretation:

1. We added the Figure presented above as Supplementary Fig. 14.
2. We refer to it in the results section as follows:

“We artificially altered exonic allele frequencies to search for any impact on other annotation features, and found modest enrichment that serves as a note of caution (Supplementary Fig. 14).”

3. We raise it again in the discussion:

“A final challenge arises from linkage disequilibrium and the overlapping nature of some genome annotation features, which together blur the relative contribution of physically co-located genomic features (Supplementary Fig. 14).”

The site by site analysis is not independent of linkage disequilibrium, sites showing high divergence in allele frequencies may just be neutral sites in high linkage disequilibrium with a true causative mutation. This is the basis of "hitchhiking". Sampling error (particularly with the number of animals used here) can cause the neutral site to show more apparent divergence than the true site, unfortunately.

Authors Response:

We feel this relates to the point above and has been addressed.

Specific comments.

Page 5 line 85. Identified not identifies

Authors' Response: Change implemented.

Page 5 line 88. Collected not collect

Authors' Response: Change implemented.

Page 6 Line 110. Replace and with however

Authors' Response: Change implemented.

Page 8. Line 167. “Two prominent examples include dorsal/ventral patterning and regulation of lipid metabolism (Table 2, Supplementary Table 7). Reduction in body size was a quick and early response to domestication across species, likely followed by human mediated regulation of muscularity and fatness as sheep were actively managed as a food source”. It is not obvious how dorsal/ventral patterning is related to the sentence that follows it, please make this clear for the reader.

Authors' Response:

We agree the use of language required improvement. We have edited the text to uncouple the examples (body patterning and metabolism), which are now presented sequentially to simplify interpretation.

Page 9. Line 199. “A large number of predicted promoters were not present, possibly reflecting our ChIP-Seq data originates from only one sheep tissue and time point”. Add “or alternatively, a large number of false positive hits”.

Authors' Response:

Our interpretation remains unchanged, as the most likely explanation remains unidentified elements given our data derived from a single tissue.

Page 12. Line 274. "Only a very small fraction of loci exhibited strongly divergent allele frequency expected to be characteristic of sites underlying domestication and/or selection". A very small fraction of loci would be expected by chance to have highly divergent allele frequencies, as a result of drift alone. The authors could simulate the divergence of their populations, (using Fregene for example), and investigate if the proportion of loci showing strongly divergent allele frequency is greater than expected by chance (drift) alone.

Authors' Response:

The assertion that SNP with divergent allele frequency between populations are characteristic of sites underlying selection underpins the basis of metrics such as F_{ST} and does not require simulation to demonstrate. We elected not to implement the reviewer's suggestion, which we felt was not a direct request. We added a relevant reference, which cites previous application of the approach used.

REVIEWERS' COMMENTS:

Reviewer #3 (Remarks to the Author):

The authors have responded satisfactorily to all previous comments, except one:

"Page 12. Line 274. "Only a very small fraction of loci exhibited strongly divergent allele frequency expected to be characteristic of sites underlying domestication and/or selection". A very small fraction of loci would be expected by chance to have highly divergent allele frequencies, as a result of drift alone. The authors could simulate the divergence of their populations, (using Fregene for example), and investigate if the proportion of loci showing strongly divergent allele frequency is greater than expected by chance (drift) alone.

Authors' Response:

The assertion that SNP with divergent allele frequency between populations are characteristic of sites underlying selection underpins the basis of metrics such as F_{ST} and does not require simulation to demonstrate. We elected not to implement the reviewer's suggestion, which we felt was not a direct request. We added a relevant reference, which cites previous application of the approach used.

It is not a satisfactory response to quote other studies which do not assess the chance of observing extreme F_{ST} values by chance alone. The authors should at the very least clearly point out that some of their results may be explained by drift between the populations rather than selection.

November 15 2017 – Final Revision

Dear Editor,

Please find attached our final revision, responding to the last request of Reviewer#3.

Best Regards,

James

Reviewer #3 (Remarks to the Author):

The authors have responded satisfactorily to all previous comments, except one:

"Page 12. Line 274. "Only a very small fraction of loci exhibited strongly divergent allele frequency expected to be characteristic of sites underlying domestication and/or selection". A very small fraction of loci would be expected by chance to have highly divergent allele frequencies, as a result of drift alone. The authors could simulate the divergence of their populations, (using Fregene for example), and investigate if the proportion of loci showing strongly divergent allele frequency is greater than expected by chance (drift) alone.

Authors' Response:

The assertion that SNP with divergent allele frequency between populations are characteristic of sites underlying selection underpins the basis of metrics such as F_{ST} and does not require simulation to demonstrate. We elected not to implement the reviewer's suggestion, which we felt was not a direct request. We added a relevant reference, which cites previous application of the approach used.

It is not a satisfactory response to quote other studies which do not assess the chance of observing extreme F_{ST} values by chance alone. The authors should at the very least clearly point out that some of their results may be explained by drift between the populations rather than selection.

Authors' Response:

We have modified the text to soften our interpretation and explicitly indicate to the reader that genetic drift is an alternative explanation. The modified text reads:

"Only a very small fraction of loci exhibited strongly divergent allele frequency. These are characteristic of sites under selection, however they can be difficult to distinguish from those impacted by genetic drift alone."